# CODER AS EDITOR: CODE-DRIVEN INTERPRETABLE MOLECULAR OPTIMIZATION

## ABSTRACT

Molecular optimization is a central task in drug discovery that requires precise structural reasoning and domain knowledge. While large language models (LLMs) have shown promise in generating high-level editing intentions in natural language, they often struggle to faithfully execute these modifications—particularly when operating on non-intuitive representations like SMILES. We introduce MECo, a framework that bridges reasoning and execution by translating editing actions into executable code. MECo reformulates molecular optimization for LLMs as a cascaded framework: generating human-interpretable editing intentions from a molecule and property goal, followed by translating those intentions into executable structural edits via code generation. Our approach achieves over 98% accuracy in reproducing held-out realistic edits derived from chemical reactions and target-specific compound pairs. On downstream optimization benchmarks spanning physicochemical properties and target activities, MECo substantially improves consistency by 38-86 percentage points to 90%+ and achieves higher success rates over SMILES-based baselines while preserving structural similarity. By aligning intention with execution, MECo enables consistent, controllable and interpretable molecular design, laying the foundation for high-fidelity feedback loops and collaborative human–AI workflows in drug discovery.

## 1 INTRODUCTION

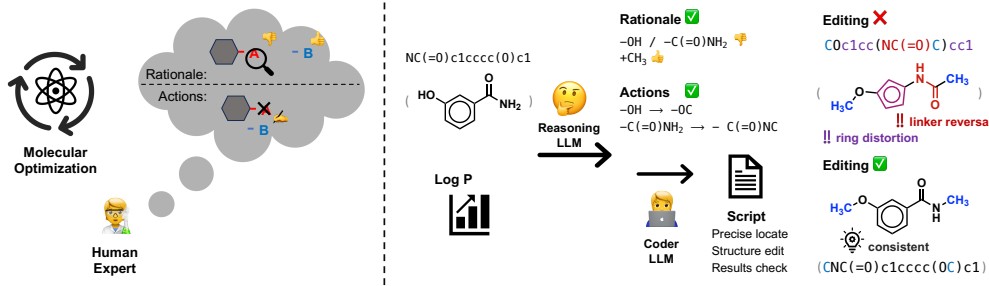

Figure 1: Motivation for MECo: bridging reasoning and execution in molecular optimization through code generation. Left: Human chemists reason over molecular graphs to design targeted edits, such as modifying a functional group to adjust polarity or introducing new interactions, and annotate them directly on the structure. Top right: Reasoning LLMs can generate similar high-level rationales and editing actions, but often struggle to execute them faithfully due to the limitations of sequential molecular representations like SMILES. Bottom right: MECo addresses this gap by introducing a coder LLM that translates editing actions into executable code, enabling precise, interpretable, and reproducible structural edits on the molecular graph.

Recent reasoning-centric large language models (LLMs) have demonstrated notable progress in scientific problem-solving, particularly in programming (Wang et al., 2023), mathematics (Lewkowycz et al., 2022), and chemistry (Zhang et al., 2024). In chemistry, their success on question answer-

ing tasks highlights an ability to encode domain knowledge (Hou et al., 2025), but applying this knowledge to tasks like molecular optimization, which require both chemical reasoning and precise structural control, remains a challenge.

A growing number of studies have applied LLMs to molecular optimization by prompting them to generate optimized molecules in the form of SMILES strings, conditioned on an input structure and a desired property goal (e.g., increased solubility or binding affinity) (Liu et al., 2024a; Ye et al., 2025). While intuitive, this generation paradigm faces major limitations. First, generated SMILES are often invalid or chemically implausible. Second, even valid outputs may diverge from the design rationale articulated by reasoning LLMs, undermining interpretability and expert trust. Such misalignments hinder human-in-the-loop workflows where reproducibility and verifiability are critical, and break the feedback loop required for iterative model refinement. Moreover, uncontrolled edits often lead to molecules that are difficult to synthesize, as they require redesigning synthetic routes and prevent reuse of intermediates, posing practical challenges and costs for experimental validation.

At the root of these limitations is a modality mismatch: SMILES is a linearized encoding of molecular graphs, originally designed for compact storage in cheminformatics systems. A single molecule may correspond to many valid SMILES depending on the traversal path, and even small structural edits can cause large, unintuitive changes in the string. In contrast, chemists typically reason over molecular graphs and use graphical tools instead of directly modifying SMILES, making precise control difficult for LLMs.

To bridge this gap, we propose using code as an intermediate representation, a domain where LLMs have already demonstrated strong proficiency. Rather than generating molecular structures directly as SMILES, we reformulate molecular editing as a code generation task: the LLM produces executable scripts (e.g., using RDKit (Landrum, 2013)) that specify verifiable and interpretable structural modifications. This approach leverages the strengths of LLMs and enables faithful execution of editing intentions to improve controllability, reproducibility, and transparency in molecular design.

We introduce **MECo** (**M**olecular **E**diting via **Co**de generation), a framework that translates high-level design rationales into structured editing programs. We fine-tune a Qwen2.5-Coder (Hui et al., 2024) model solely on synthetic data generated by limited moiety substitutions on random molecules. The model achieves 98% accuracy on realistic edits derived from reaction and bioactivity datasets, significantly outperforming a general Qwen2.5 (Yang et al., 2025) model fine-tuned to directly generate SMILES. When using a reasoning LLM such as Deepseek-R1 (Guo et al., 2025) as the upstream component, MECo substantially improves the consistency between editing intentions and resulting structures by 38-86 percentage points to 90%+, leading to higher success rates while preserving structural similarity across multiple property and activity optimization benchmarks.

By bridging natural language reasoning and structural molecular modification through code, MECo enables consistent, controllable and interpretable molecular optimization, bringing LLM-guided molecule design closer to real-world scientific application. Our main contributions are:

- **Code-based formulation for molecular optimization.** We introduce MECo, a novel framework that recasts molecular editing as a code generation task, enabling LLMs to translate natural language intentions into verifiable and executable structural modifications.

- **Scalable data construction for training and evaluation.** We develop a scalable pipeline for constructing both synthetic and realistic editing samples, combining programmatic moiety replacement with edit extraction from chemical reactions and bioactive molecule pairs.

- **Generalization to realistic molecular transformations.** We show that a code LLM trained solely on synthetic edits generalizes effectively to real-world modifications, achieving over 98% accuracy on both reaction- and activity-derived edits.

- **Superior molecule optimization performance.** MECo outperforms direct generation baselines across property and activity benchmarks, with notably *double the structure–intention consistency*, enhancing interpretability and reliability.

## 2    RELATED WORK

Among various molecular representations, SMILES (Weininger, 1988) has been widely adopted in sequence-based models, including RNNs (Gómez-Bombarelli et al., 2018; Segler et al., 2018) and

Transformers (Schwaller et al., 2019). In contrast, graph-based approaches (Gilmer et al., 2017; Jin et al., 2018a; Shi et al., 2020) and 3D-aware models (Schütt et al., 2017; Satorras et al., 2021) aim to better capture structural validity by operating directly on molecular graphs and spatial coordinates. Large-scale pretraining further produced molecular language models such as ChemBERTa (Chithrananda et al., 2020), MolBERT (Fabian et al., 2020), Chemformer (Irwin et al., 2022), and MolXPT (Liu et al., 2023), which perform competitively with graph-based approaches. However, SMILES poorly align with natural language: small structural edits can cause large string differences, and multiple encodings exist for the same molecule (Merz Jr et al., 2020), making them suboptimal for reasoning in LLMs.

Graph-based molecular optimization has been extensively studied, including gradient-based optimization (Jin et al., 2018a), RL-based editing (Jin et al., 2020b; Shi et al., 2020),, low-to-high motif translation (Jin et al., 2018b; 2020a), and guided diffusion (Vignac et al., 2022). These approaches are powerful but fundamentally rely on oracle-driven optimization(Gao et al., 2022), using predictors, learned surrogates, or reward signals to characterize the underlying property landscape. More recently, reasoning LLMs have introduced a complementary, zero-oracle perspective, where edits can be proposed directly from embedded chemical knowledge without querying task-specific oracles. This opens opportunities for scenarios where oracles are unavailable (e.g., new targets) or expensive (e.g., experimental endpoints), and can naturally interface with existing oracle-based optimizers or language–graph alignment approaches to further expand the molecular design space.

LLMs have been investigated as general-purpose optimizers (Yang et al., 2023; Meyerson et al., 2024; Liu et al., 2024b), and these ideas have recently been extended to molecular domains. Prompt-based approaches such as MOLLEO (Wang et al., 2024) and ChatDrug (Liu et al., 2024a) adapt LLMs to propose molecular modifications, either by embedding them in genetic algorithms or by augmenting them with retrieval databases. Other methods rely on representation learning, for example by exploiting pretrained LLM embeddings (Ranković & Schwaller, 2023) or by fine-tuning general-purpose models on molecular corpora to improve generation quality (Bedrosian et al., 2024; Fang et al., 2023; Kristiadi et al., 2024). DrugAssist (Ye et al., 2025) further contributed a large-scale MolOpt-Instructions dataset to support instruction-tuned optimization models. LICO (Nguyen & Grover, 2024) proposed a semi-synthetic training framework that extends general-purpose LLMs into surrogate models for black-box molecular optimization. MolReasoner (Zhao et al., 2025) introduces a two-stage framework that integrates synthetic Chain-of-Thought supervision with reinforcement learning, shifting molecular LLMs from memorization toward interpretable reasoning. MultiMol (Yu et al., 2025) further improves LLM-based molecular optimization through a multi-agent framework that specializes individual LLM agents for different subtasks and retrieves relevant external literature and data. Despite these advances, most existing systems still operate directly in SMILES or graph spaces, which limits their alignment with natural language reasoning and hinders the interpretability of the generated modifications.

## 3 METHODS

### 3.1 PROBLEM FORMULATION

Molecular optimization is a central task in drug discovery, where the objective is to generate a modified molecule $M_o$ from an initial compound $M_i$ to improve one or more target properties $T$ (e.g., permeability, target binding affinity), while preserving essential structural features such as the core scaffold or pharmacophores, and enabling reuse of steps in a common synthetic route.

In many prior approaches, particularly those based on RNNs or early LLMs, this task is formulated as:

$$M_o = \mathcal{F}(M_i, T) \tag{1}$$

where $\mathcal{F}$ is a black-box model that directly maps the input molecule and target to a SMILES string.

While this formulation has shown some empirical success, it suffers from two key limitations. First, the editing process is entirely implicit: the model does not explain what was changed or why, making the transformation uninterpretable. Second, the output often diverges from well-established medicinal chemistry principles. These models tend to make broad, unconstrained modifications, rather than the minimal, targeted edits that chemists use, such as modifying a functional group to adjust polarity for better permeability, or introducing an H-bond donor to improve affinity and selectivity.

Without such structure–property reasoning, the results are hard to interpret or trust, limiting their practical utility in real-world design workflows.

In contrast, recent reasoning-centric LLMs offer a fundamentally different capability: rather than directly generating $M_o$, they can first articulate a structured editing intention based on $M_i$ and $T$. These intentions often consist of: **a set of editing actions** $A = \{a_1, a_2, \dots\}$ (e.g., "replace the para-methyl with a hydroxyl group"), and **a corresponding rationale** (e.g., "to increase polarity and improve solubility").

This shift from black-box generation to interpretable, rationale-driven editing actions more closely reflects how medicinal chemists reason about molecular design based on structural and physico-chemical considerations. The corresponding formulation becomes:

$$(A, M_o) \leftarrow \mathcal{F}(M_i, T) \tag{2}$$

However, in current systems, this promising capability remains underutilized. While reasoning LLMs can propose chemically meaningful edits, they often fail to execute them faithfully. This is largely due to the difficulty of performing precise and constrained modifications on SMILES representations, which are sensitive to minor syntax errors and lack structural locality. As a result, the generated molecule $M_o$ may deviate from the proposed intention or violate chemical constraints.

To bridge this gap between reasoning and execution, we reformulate molecular optimization as a code generation task:

$$\mathcal{C} = \mathcal{G}(M_i, A), \quad M_o = \mathcal{C}(M_i) \tag{3}$$

where $\mathcal{G}$ denotes the code generation model (e.g., a coder LLM), and $\mathcal{C}$ is an executable script (e.g., using RDKit) that implements the specified editing actions $A_i$.

**Why code as an interface?** LLMs have demonstrated strong performance in translating natural language into structured domain-specific languages such as programming code (Fang et al., 2024). Models like CodeT5+ (Wang et al., 2023) and StarCoder (Li et al., 2023) can reliably generate and edit code given natural language instructions. This success is attributed to the syntactic and semantic regularity of code, as well as structure-aware strategies such as abstract syntax trees (ASTs) (Wang et al., 2021), fill-in-the-middle training (Li et al., 2023), and execution-based feedback (Li et al., 2022). In complex or safety-critical domains, formal specifications can be further integrated to enhance correctness and verifiability (Murphy et al., 2024).

This motivates us to treat molecular editing as a code generation task. Rather than generating fragile SMILES strings, the LLM produces executable scripts that specify structural edits. Notably, chem-informatics libraries like RDKit (Landrum, 2013) provide robust APIs for manipulating molecular graphs, allowing SMILES strings to be parsed into graph-based data structures and modified pro-grammatically. This enables interpretable, verifiable, and reproducible molecular transformations grounded in chemical structure, not string syntax.

## 3.2 Framework Overview

We propose **MECo** (**M**olecular **E**diting via **Co**de generation), a framework that decouples high-level chemical reasoning from low-level molecular editing, as illustrated in Figure 2C. Given an initial molecule $M_i$ and a target property goal $T$, a reasoning LLM first generates an editing intention $(A, \text{rationale})$, where $A$ denotes a set of editing actions and the rationale explains their purpose. The focus shifts to executing $A$ through code generation, producing the optimized molecule $M_o$.

MECo operates in a cascading manner:

**Intention generation (reasoning LLM):** Generates editing actions $A$ and associated rationales based on $(M_i, T)$.

**Action execution (code LLM):** Translates $A$ into code $\mathcal{C}$, which is applied to $M_i$ to yield $M_o$.

$$\underbrace{(M_i, T) \longrightarrow (A, \text{ rationale})}_{\text{reasoning LLM}} \longrightarrow \underbrace{\mathcal{C} = \text{CodeGen}(A)}_{\text{code LLM}} \longrightarrow M_o = \mathcal{C}(M_i) \tag{4}$$

This decoupled formulation offers several advantages: (1) It separates reasoning (*why* to edit) from execution (*how* to edit), mirroring expert design workflows. (2) It provides interpretable, verifiable,

Figure 2: Overview of the MECo framework. (A) Synthetic data construction for model training. (B) Realistic data for testing on unseen molecular modifications derived from both reaction and activity data, reflecting chemical transformations observed in practice and in human reasoning. Modifications are categorized into terminal or core replacements. (C) End-to-end workflow and evaluation of the framework. The coder LLM is evaluated both independently and within the integrated pipeline. A reasoning LLM first generates rationales and editing actions, which are then executed by the coder model to produce optimized molecules. Outputs are subsequently reviewed by human experts to assess the alignment between generated structures and intended editing actions.

and debuggable records of molecular transformation. (3) It ensures structural validity and fine-grained control by grounding edits in explicitly defined procedures.

By translating natural language intentions into code-level structural edits, MECo enable LLM-driven molecule design that is not only intelligent, but also reliable, transparent, and practically applicable to real-world drug discovery pipelines.

## 3.3 DATA CONSTRUCTION

To train the code LLM to translate editing actions into executable molecular transformations, we construct a large-scale dataset of molecule–edit–code triples in the form:

$$(M_i, A) \longrightarrow \mathcal{C}$$

where $M_i$ is the input molecule, $A$ is a set of editing actions expressed in natural language, and $\mathcal{C}$ is the corresponding Python code that applies the edit using cheminformatics toolkits such as RDKit.

We adopt a hybrid data construction strategy combining synthetic and realistic samples:

**Synthetic edits.** We constructed synthetic editing examples by applying programmatically defined moiety replacements to molecules randomly sampled from the ZINC database (Sterling & Irwin, 2015), as shown in Figure 2A. Each initial molecule $M_i$ was iteratively edited by substituting one of its substructures with a moiety from a predefined pattern pool $\mathcal{P} = \bigcup_i \mathcal{P}_i$, where each subset $\mathcal{P}_i$ contains fragments with $i$ attachment points.

At each iteration, a pattern $p$ was randomly drawn from the pool, and a matching site in $M_i$ was identified. If a match was found, it was replaced with another randomly selected moiety $r \in \mathcal{P}_i \setminus \{p\}$ of the same connectivity. If no match was found, the process continued with a different pattern until the pool was exhausted. Replacement was performed iteratively, with all successful edits recorded. For each edit, the corresponding executable code snippet $\mathcal{C}$ was automatically generated, and the edited molecule $M_o$ was obtained by applying $\mathcal{C}$ to the input molecule $M_i$, i.e., $M_o = \mathcal{C}(M_i)$.

This procedure yields chemically valid and structurally controlled synthetic edits, providing precise edit-code supervision that enables the model to learn robust mappings from actions to code without requiring exposure to real-world editing data. The full algorithm and list of predefined patterns are provided in Appendix A.1.

**Realistic edits.** To evaluate generalization beyond synthetic edits, we construct a set of realistic editing samples derived from two sources (Figure 2B):

**Reaction-derived edits:** We extracted matched reactant–product pairs from the USPTO-MIT reaction dataset (Jin et al., 2017). Atom mapping was used to identify structural changes, and we retained only samples with a single modification site while preserving the shared molecular core.

**Target-based edits:** We extracted compounds from the ChEMBL35 database (Mendez et al., 2018) and grouped them by target ID after applying basic structure-based filters (details provided in Appendix A.2). Matched molecular pairs (MMPs) were identified using the algorithm by Hussain & Rea (2010), and further filtered following the criteria described in Appendix A.3. Attachment points were explicitly identified and recorded to enable precise modification tracking. Based on the number of attachment points in each transformation, the data were categorized into two types: **terminal replacements** (single-point attachment, same as those in **Reaction-derived edits**) and **core replacements** (multi-point attachment).

Together, these two sources reflect the two most common strategies in real-world molecular design. The first captures reaction-derived transformations, grounded in feasible chemical synthesis steps. The second comprises target-specific structural modifications observed across bioactive compounds, which represent plausible edits made during lead optimization. While the directionality of property change is not explicitly defined in our setup, these samples provide chemically meaningful, human-curated edits. Although our model is trained solely on synthetic data, evaluation on these realistic edits allows us to assess its generalization to authentic molecular transformation scenarios.

### 3.4 MODEL TRAINING

To train the coder LLM for accurate molecular editing, we used a curated dataset of 50,000 synthetic editing samples (Section 3.3). Each sample includes a molecule $M_i$, an editing action $A$ described using a combination of natural language and cheminformatics notation (e.g., SMARTS), and a Python code snippet $\mathcal{C}$ that transforms $M_i$ accordingly. For training the direct SMILES generation baseline, the corresponding output molecules $M_o = \mathcal{C}(M_i)$ were used as targets. All models were fine-tuned using the official Qwen2.5-Coder finetuning framework. Additional details regarding prompt design, formatting choices, and training settings are provided in Appendix B.

### 3.5 BENCHMARK AND EVALUATION METRICS

To evaluate both the core editing capability and downstream utility of our framework, we introduced two complementary benchmarks aligned with our problem formulation:

**1. Molecular editing benchmark for problem diagnosis.** We used the realistic editing samples constructed in Section 3.3 to identify the limitations of existing approaches and to quantitatively evaluate the molecular editing capabilities of MECo. For each sample, we assess whether the model can faithfully execute the specified editing action $A$ on the input molecule $M_i$, either directly or via code generation, to produce the expected output molecule $M_o$.

**2. Molecular optimization benchmark for application endpoint.** To assess the practical utility of our framework, we evaluate end-to-end performance on ChemCoTBench (Li et al., 2025a). This benchmark comprises six molecular optimization tasks: three involving physicochemical properties (penalized logP (Gómez-Bombarelli et al., 2018), solubility (Delaney, 2004), QED (Bickerton et al., 2012), implemented in RDKit (Landrum, 2013)) and three involving target-specific bioactivities (DRD2 (Olivecrona et al., 2017), JNK3, and GSK3$\beta$ (Li et al., 2018), implemented in TDC Oracles (Huang et al., 2021)). Each task provides a set of input molecules $M_i$ paired with a shared optimization goal $T$. This setting reflects real-world design scenarios, where high-level property goals must be translated into concrete molecular modifications. For each $(M_i, T)$ pair, we compare

direct molecule generation from a reasoning LLM with the MECo framework, which first generates an editing intention and then executes it via code. This evaluation allows us to assess whether MECo improves over direct generation approaches in terms of (1) chemical validity, (2) optimization success rate, (3) mean similarity to source molecules, and (4) structure–action consistency, which is computed only for cases with human-interpretable edits. We omit mean property improvement from the main results, since models can achieve large gains by sacrificing structural preservation and producing molecules that differ substantially from the initial ones. Such cases inflate the average improvement without reflecting meaningful optimization. We therefore adopt success rate, and provide the mean improvement results in Appendix C.5.

## 4 EXPERIMENTS

### 4.1 EXPERIMENTAL SETUP

In the molecular optimization task under the MECo framework, we applied a unified wrapper to the original prompt to constrain the output format and facilitate reliable extraction of editing actions. The full wrapper template is provided in Appendix C.2. To obtain ground-truth labels, we manually applied the actions generated by the reasoning LLMs to the initial molecules, following the evaluation criteria described in Appendix C.3.

We selected DeepSeek-R1 (Guo et al., 2025) as the reasoning LLM used in MECo by default, due to its strong performance in structured scientific reasoning and its fully open-source availability, which facilitates reproducibility and downstream integration. Gemini-2.5-Pro (Comanici et al., 2025) was also evaluated for comparison, given its top performance on ChemCoTBench. However, its internal reasoning mechanism is proprietary, and it remains uncertain whether external tools are invoked during inference, especially considering its multi-modal and mixture-of-experts (MoE) architecture.

### 4.2 EDIT EXECUTION ON REALISTIC BENCHMARK

To ensure a fair evaluation of editing generalization, we removed all samples from each realistic dataset whose original or modified fragments had a Tanimoto similarity $\geq 0.6$ (computed using ECFP4 (Rogers & Hahn, 2010)) with any moiety used in the synthetic training set. From the remaining pool, we randomly sampled 1,000 examples per category: reaction-derived, target-specific terminal replacement, and target-specific core replacement, as the test sets. This setup ensures that the model is evaluated on structurally diverse and unseen edits, providing a robust assessment of its generalization ability. Results are summarized in Table 1. Appendix A.5 further visualizes the fragment distributions via t-SNE, showing the broad and largely non-overlapping chemical space of realistic edits.

Table 1: Execution accuracy (%) across realistic edit benchmarks. Rows are shaded to distinguish model variants: Qwen2.5 , Qwen2.5-Coder , and finetuned versions (darker, denoted by -FT).

| Action | Model | Terminal replacement | | Core replacement |
| --- | --- | --- | --- | --- |
| | | Reaction-derived | Target-specific | Target-specific |
| Direct | Qwen2.5-7B-Instruct | 0.5 | 1.3 | 0.0 |
| | Qwen2.5-7B-Instruct-FT | 38.7 | 7.4 | 2.7 |
| CodeGen | Qwen2.5-7B-Instruct | 35.4 | 3.7 | 4.1 |
| | Qwen2.5-Coder-7B-Instruct | 50.3 | 9.9 | 12.2 |
| | Qwen2.5-Coder-7B-Instruct-FT (ours) | **99.9** | **98.3** | **98.3** |

This comparison demonstrates the advantage of code-based molecular editing over direct SMILES generation. The direct generation model Qwen2.5-7B-Instruct exhibits extremely low execution accuracy across all categories, suggesting a limited ability to faithfully apply structural modifications. Notably, even without specialized training, prompting the same general LLM to generate code leads to substantially higher accuracy than its direct generation counterpart, underscoring the inherent advantages of structured code over direct SMILES generation. Interestingly, we find that the discrepancy between reaction-derived and target-specific terminal replacements can be largely attributed to differences in SMILES syntax patterns, suggesting that LLMs struggle to generalize

over SMILES syntax in both direct generation and code generation. A detailed discussion is provided in Appendix C.4.

With supervised fine-tuning, the code-specialized model Qwen2.5-Coder-7B-Instruct-FT achieves over 98% execution accuracy across all benchmarks, while direct generation shows only marginal improvement. These results affirm that MECo's code generation formulation dramatically improves execution fidelity, even on realistic and diverse editing tasks.

### 4.3 IMPROVEMENT ON MOLECULAR OPTIMIZATION

Table 2 summarizes performance across six molecular optimization tasks, covering both physicochemical properties and target activities. We compare direct SMILES generation from several non-reasoning / reasoning LLMs with our proposed MECo framework, which augments reasoning-centric molecular optimization with code generation for editing execution.

MECo improves success rates (SR%) while maintaining higher structural similarity (Sim), suggesting more effective yet conservative edits. Importantly, it shows markedly higher consistency rate (CR%), which measure alignment between output structures and editing intentions, highlighting its faithfulness to reasoning rationales. This alignment enhances interpretability by ensuring that the optimized molecules reflect the reasoning LLM's intended actions, resulting in more consistent and transparent optimization that can be readily understood and verified by human experts.

In contrast, direct SMILES generation often produces unintended or overly disruptive modifications, with poor alignment to the specified editing instructions (see cases in Section 4.4). Even strong reasoning LLMs fall short when lacking explicit action execution, underscoring the need to bridge high-level reasoning and low-level structural manipulation through code-based editing.

Figure 3 further shows that MECo's improvements generalize across reasoning LLMs (DeepSeek-R1 and Gemini-2.5-Pro), demonstrating its broad applicability and utility in controllable, interpretable molecule design.

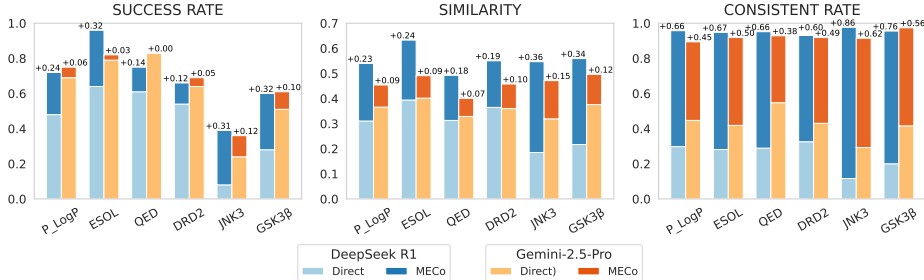

Figure 3: Incremental bar plots comparing direct generation (light bars) and MECo (dark stacked bars) across tasks and metrics under two different reasoning LLMs. Numbers above the bars indicate the relative improvements of MECo over direct.

### 4.4 CASE STUDY

To illustrate the difference between direct generation and our MECo framework, we analyze molecular optimization tasks and present representative cases in Figure 4A. Direct generation often introduces uncontrolled structural changes with low similarity to the initial molecule, leading to unrealistic score inflation or failed optimization. In contrast, MECo achieves improved scores while maintaining high similarity through interpretable, chemistry-preserving edits.

The detailed example on GSK3$\beta$ activity optimization task in Figure 4B further demonstrates how MECo maintains consistency across the reasoning–execution–analysis pipeline. The reasoning LLM proposes rational modifications, such as replacing a hydroxy with an amino to strengthen hydrogen bonding and improve solubility, and substituting a phenyl ring with a pyridin-3-yl ring to introduce an additional hydrogen bond acceptor and enhance pharmacokinetic potential. The coder LLM then faithfully translates these actions into executable edits, ensuring that the intended modifications are

Table 2: Performance on molecular optimization tasks. Each task reports validity rate (VR%), success rate (SR%), mean similarity to the source molecule (Sim), and consistency rate (CR%). **Bold** numbers highlight the best value in each column. Note: GPT-5 refused to answer a small number of samples.

| | **Property optimization tasks** | | | | | | | | | | | |
| **Model** | **Penalized logP** | | | | **Solubility** | | | | **QED** | | | |
| | VR% | SR% | Sim | CR% | VR% | SR% | Sim | CR% | VR% | SR% | Sim | CR% |
| *W/o thinking* | | | | | | | | | | | | |
| GPT-4o | 22 | 8 | 0.11 | – | 41 | 35 | 0.21 | – | 31 | 17 | 0.11 | – |
| Gemini-2.0-flash | 66 | 44 | 0.27 | – | 32 | 28 | 0.13 | – | 64 | 54 | 0.26 | – |
| DeepSeek-V3 | 53 | 21 | 0.20 | – | 52 | 44 | 0.21 | – | 48 | 33 | 0.18 | – |
| *W/ thinking* | | | | | | | | | | | | |
| GPT-5 | 58 | 32 | 0.21 | – | 67 | 63 | 0.25 | – | 68 | 66 | 0.21 | – |
| Gemini-2.5-Pro | 78 | 69 | 0.37 | 45 | 79 | 79 | 0.40 | 42 | 83 | **83** | 0.33 | 55 |
| DeepSeek-R1 | 62 | 48 | 0.31 | 30 | 70 | 64 | 0.39 | 28 | 68 | 61 | 0.31 | 29 |
| **MECo** | **93** | **72** | **0.54** | **96** | **96** | **96** | **0.63** | **95** | **87** | 75 | **0.49** | **95** |
| | **Activity optimization tasks** | | | | | | | | | | | |
| **Model** | **DRD2** | | | | **JNK3** | | | | **GSK3$\beta$** | | | |
| | VR% | SR% | Sim | CR% | VR% | SR% | Sim | CR% | VR% | SR% | Sim | CR% |
| *W/o thinking* | | | | | | | | | | | | |
| GPT-4o | 27 | 12 | 0.12 | – | 20 | 4 | 0.08 | – | 19 | 10 | 0.08 | – |
| Gemini-2.0-flash | 60 | 32 | 0.29 | – | 56 | 17 | 0.25 | – | 71 | 39 | 0.36 | – |
| DeepSeek-V3 | 46 | 22 | 0.20 | – | 40 | 9 | 0.14 | – | 43 | 18 | 0.15 | – |
| *W/ thinking* | | | | | | | | | | | | |
| GPT-5 | 64 | 50 | 0.20 | – | 57 | 15 | 0.17 | – | 46 | 28 | 0.14 | – |
| Gemini-2.5-Pro | 85 | 64 | 0.36 | 43 | 74 | 24 | 0.32 | 42 | 74 | 51 | 0.38 | 29 |
| DeepSeek-R1 | 73 | 54 | 0.36 | 33 | 43 | 8 | 0.19 | 12 | 45 | 28 | 0.22 | 20 |
| **MECo** | **89** | **66** | **0.55** | **93** | **92** | **39** | **0.55** | **98** | **91** | **60** | **0.56** | **96** |

precisely applied. This yields improved oracle and docking scores, while complementary binding analysis further validates the rationales. The coherent consistency underpins the interpretability and reliability of MECo compared to direct generation.

## 5 CONCLUSION

We introduce MECo, a framework that reformulates molecular optimization as a code generation task, bridging high-level reasoning with low-level structural execution. By translating interpretable editing intentions into executable scripts, MECo achieves near-perfect edit fidelity on realistic benchmarks and consistently outperforms direct-SMILES-generation baselines across both physicochemical property and bioactivity optimization tasks. This formulation enables consistent, controllable and interpretable molecular optimization workflows, paving the way for trustworthy human-AI collaboration and high-fidelity feedback loops between AI models and experimental validation in drug discovery, such as agent-style extensions and multi-round optimization. Beyond this application, our results suggest a general paradigm for bridging reasoning and execution in scientific domains, highlighting the potential of LLMs as reliable assistants for structured, verifiable discovery workflows.

**Ethics statement.** Our work focuses on developing a framework for interpretable molecular optimization using large language models. The study does not involve human subjects, personally identifiable data, or sensitive demographic attributes. The molecular datasets used are publicly available or derived from standard cheminformatics benchmarks, and no proprietary or confidential information is disclosed. The methods are designed for research purposes in computational drug discovery and do not directly produce clinically actionable compounds without further experimental valida-

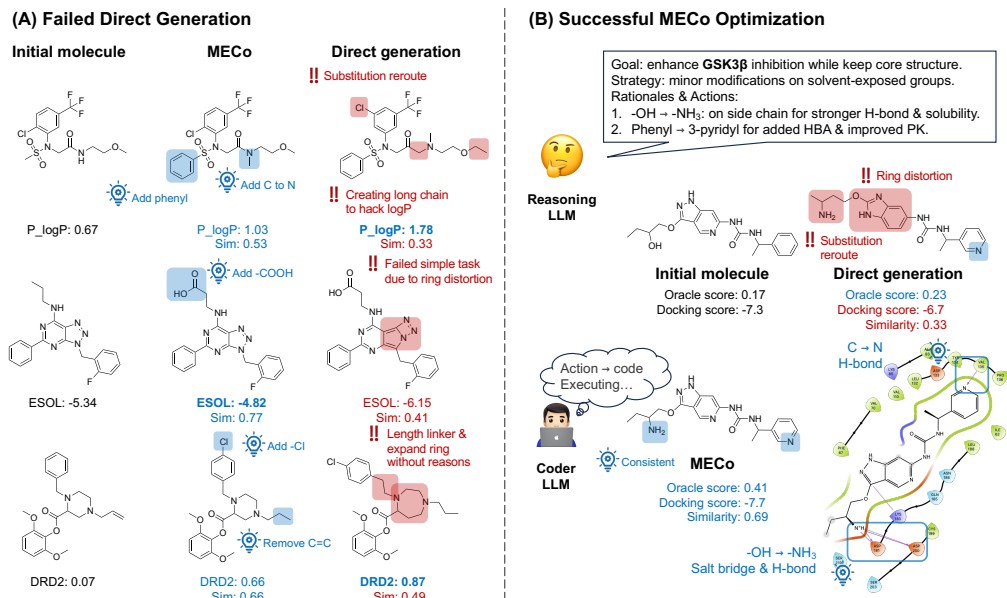

Figure 4: Representative case study of molecular optimization. (A) Direct generation produces uncontrolled changes and low similarity, often leading to unrealistic score inflation or failed optimization; MECo achieves improved scores and higher similarity through interpretable, chemistry-preserving edits. (B) MECo demonstrates coherent consistency across the reasoning–execution–analysis pipeline.

tion. We see no ethical concerns beyond the general risks associated with generative molecular design, which we mitigate by emphasizing interpretability.

**Reproducibility statement.** We have made efforts to ensure reproducibility by documenting data construction, model settings, and evaluation protocols, and will release code, model parameters, and processed datasets upon publication. For API-based LLMs, we acknowledge a degree of uncontrollability due to potential changes in availability or provider-specific differences. To reduce randomness, we fixed the temperature to 0 and top_p to 1, though minor fluctuations in metrics may still occur. Outputs from locally deployed models (Qwen2.5 and Qwen2.5-Coder) are fully deterministic. For the manual evaluation, some subjectivity is unavoidable, but we followed the criteria provided in the appendix to ensure consistency and fairness.

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

# A  DATA CONSTRUCTION DETAILS

## A.1  SYNTHETIC EDIT GENERATION ALGORITHM

We provide here the full algorithm for generating synthetic editing examples, as described in Section 3.3. The procedure applies pre-defined moiety replacements to randomly sampled molecules from the ZINC database, using structured pattern pools grouped by attachment point count.

---

**Algorithm 1** Iterative Moiety Replacement

---

**Require:** Molecule $M_i$, pattern pool $\mathcal{P} = \bigcup_i \mathcal{P}_i$, iterations $N$
1: $M \leftarrow M_i$ {Initialize molecule}
2: **for** $n = 1$ to $N$ **do**
3:     $\mathcal{P}_{\text{avail}} \leftarrow \mathcal{P}$ {Reset available patterns}
4:     $replaced \leftarrow$ False
5:     **while** $\mathcal{P}_{\text{avail}} \neq \emptyset$ and $replaced =$ False **do**
6:         Sample $p \sim \mathcal{P}_{\text{avail}}$; $\mathcal{P}_{\text{avail}} \leftarrow \mathcal{P}_{\text{avail}} \setminus \{p\}$ {Draw and remove pattern}
7:         $\mathcal{S} \leftarrow$ GETSUBSTRUCTMATCHES$(M, p)$ {Identify all matching substructures}
8:         **if** $\mathcal{S} \neq \emptyset$ **then**
9:             $s \sim \mathcal{S}$ {Randomly select a match site}
10:          $i \leftarrow$ attachment_count$(p)$ {Determine connection type}
11:          $r \sim \mathcal{P}_i \setminus \{p\}$ {Sample replacement with same connectivity}
12:          **for** $a$ in $M$ **do**
13:            **if** $a \notin s$ **then**
14:              Protect$(a)$ {Protect atoms not in the selected substructure match $s$}
15:            **end if**
16:          **end for**
17:          $M \leftarrow$ replace$(M, p, r)$ {Apply replacement using REACTION}
18:          Record $(p, s, r)$ {$s$ uniquely specifies the matched substructure instance}
19:          $replaced \leftarrow$ True
20:         **end if**
21:     **end while**
22:     **if** $replaced =$ False **then**
23:         **break** {Stop further iterations if replacement failed}
24:     **end if**
25: **end for**
26: **return** $M_o$, $\{(p_j, s_j, r_j)\}_{j=1}^{n}$ {Final molecule and list of edits}

---

The lists of predefined substituents (Table 3) and molecular linkers (Table 4) are provided for constructing synthetic editing samples.

## A.2  COMPOUND PREPROCESSING AND FILTERING

We extracted standardized SMILES strings and associated bioactivity data from the ChEMBL35 database. To ensure molecular quality and downstream compatibility, we applied the following filtering criteria:

- Salt removal: only the largest fragment was retained;
- Molecular weight between 100 and 800 Da;
- Allowed atom types: H, C, N, O, F, Cl, Br, I, S, P, B, Se;
- No linear unbranched chains longer than six heavy atoms.

These filters resulted in a curated set of drug-like molecules, which served as the input for matched molecular pair (MMP) generation.

## A.3  TARGET GROUPING AND MMP ANALYSIS

After filtering, compounds were grouped by target ID, and each group was exported as a separate `.smi` file. For each group, matched molecular pairs (MMPs) were generated using the RDKit Con-

Table 3: Common substituents for synthetic data.

| Category | Name | SMILES |
|---|---|---|
| Halogens | Fluoro | [*:1]F |
| | Chloro | [*:1]Cl |
| | Bromo | [*:1]Br |
| | Iodo | [*:1]I |
| Alkyl | Methyl | [*:1]C |
| | Ethyl | [*:1]CC |
| | Isopropyl | [*:1]C(C)C |
| | tert-Butyl | [*:1]C(C)(C)C |
| Aryl | Phenyl | [*:1]c1ccccc1 |
| | p-Tolyl | [*:1]c1ccc(cc1)C |
| | p-Chlorophenyl | [*:1]c1ccc(cc1)Cl |
| Oxygen-containing | Hydroxyl | [*:1]O |
| | Methoxy | [*:1]OC |
| | Ethoxy | [*:1]OCC |
| | Carboxyl | [*:1]C(=O)O |
| | Aldehyde | [*:1]C=O |
| | Ketone | [*:1]C(=O)C |
| Nitrogen-containing | Amino | [*:1]N |
| | Methylamino | [*:1]NC |
| | Dimethylamino | [*:1]N(C)C |
| | Cyano | [*:1]C#N |
| | Nitro | [*:1][N+](=O)[O-] |
| Sulfur-containing | Thiol | [*:1]S |
| | Methylthio | [*:1]SC |
| | Sulfonyl | [*:1]S(=O)(=O)C |

trib MMPA pipeline (`rfrag.py` and `indexing.py`). This procedure applies systematic bond fragmentation and indexing to identify molecular pairs that differ by a single localized transformation. Each transformation was encoded as a SMIRKS pattern, representing the minimal structural change. All MMP results were merged and formed a high-quality pool of target-specific matched molecular pairs.

### A.4 TRANSFORMATION TYPE CLASSIFICATION

To better characterize the nature of molecular edits, the resulting MMPs were further categorized into two types based on structural properties:

**1. Core replacement.** MMPs were classified as core replacements if they satisfied the following criteria:

- **Attachment point constraint:** both fragments have the same number of attachment points, with count $\geq 2$.

- **Scaffold difference:** the Murcko scaffolds of the prior and latter fragments in replacement are different;

- **Ring requirement:** both fragments contain at least one ring;

- **Non-ring atom constraint:** each fragment contains no more than 5 non-ring heavy atoms, including the attachment points, to avoid heavily decorated rings;

- **Component ratio:** the core fragment in the initial molecule account for less than 50% heavy atoms.

Table 4: Common molecular linkers for synthetic data.

| Category | Name | SMILES |
|---|---|---|
| Aromatic linkers | Meta-phenylene | [*:1]c1cc([*:2])ccc1 |
| | Para-phenylene | [*:1]c1ccc([*:2])cc1 |
| | Ortho-phenylene | [*:1]c1c([*:2])cccc1 |
| Carbonyl-based linkers | Amide | [*:1][C;!R](=O)[N;!R][*:2] |
| | Reverse amide | [*:1][N;!R][C;!R](=O)[*:2] |
| | Ester | [*:1][C;!R](=O)[O;!R][*:2] |
| | Ketone bridge | [*:1][C;!R](=O)[*:2] |
| | Urea | [*:1][N;!R][C;!R](=O)[N;!R][*:2] |
| | Carbamate | [*:1][O;!R][C;!R](=O)[N;!R][*:2] |
| | Sulfonamide | [*:1]S(=O)(=O)[N;!R][*:2] |
| Alkyl / heteroatom linkers | Methylene | [*:1][C;!R][*:2] |
| | Ethylene | [*:1][C;!R][C;!R][*:2] |
| | Ether | [*:1][O;!R][*:2] |
| | Thioether | [*:1][S;!R][*:2] |
| | Secondary amine | [*:1][N;!R][*:2] |
| Extended / heterocyclic linkers | 1,2,3-Triazole | [*:1]c1nnn([*:2])c1 |
| | Imidazole-type | [*:1]c1[nH]cc([*:2])n1 |
| | Piperazine | [*:1]N1CCN([*:2])CC1 |
| | Piperidine | [*:1]N1CCC([*:2])CC1 |
| Polar chain linker | PEG unit (ethylene glycol) | [*:1][O;!R][C;!R][C;!R][O;!R][*:2] |

In cases where the original fragment exhibits symmetry and the replacement fragment does not (as illustrated in the last line of Figure 5), multiple distinct products may arise due to different permutations of attachment point mappings, accounting for 7.5% of our test set. Since the original attachment sites are chemically interchangeable, all such permutations are treated as valid ground truths for evaluation.

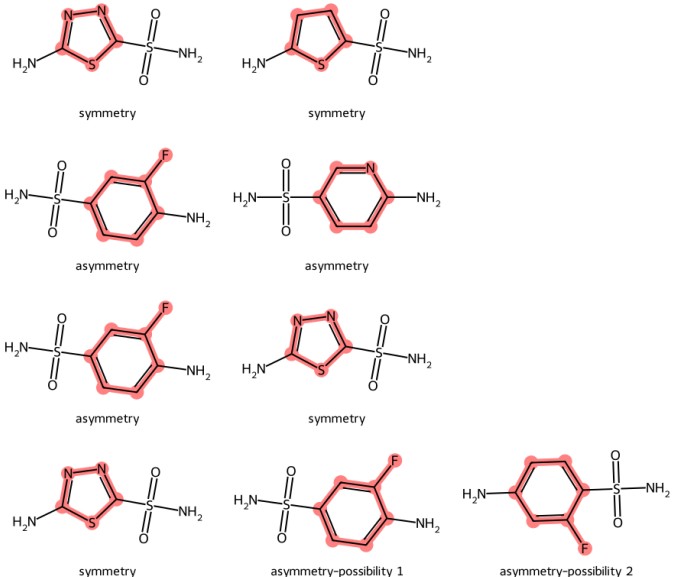

Figure 5: Multiple possible permutations in symmetry-to-asymmetry core replacement.

**2. terminal replacement.** MMPs were classified as single-point replacements (or terminal replacements) if the number of heavy atoms in the modified fragment accounts for no more than 30% of the initial molecule and only one attachment point.

## A.5 T-SNE VISUALIZATION OF FRAGMENT CHEMICAL SPACE

To assess the chemical diversity of the Realistic Edits test set relative to the synthetic moiety pool used in training, we computed ECFP fingerprints (radius = 2, 2048 bits) for all fragments and projected them into two dimensions using t-SNE. As shown in Figure 6, fragments from the synthetic training pool cover only a small portion of the chemical space and form several compact clusters, reflecting the limited structural motifs used to construct the synthetic edit pairs. In contrast, fragments from the Realistic Edits set are widely dispersed across the projection space, occupying diverse regions that are largely non-overlapping with the synthetic clusters.

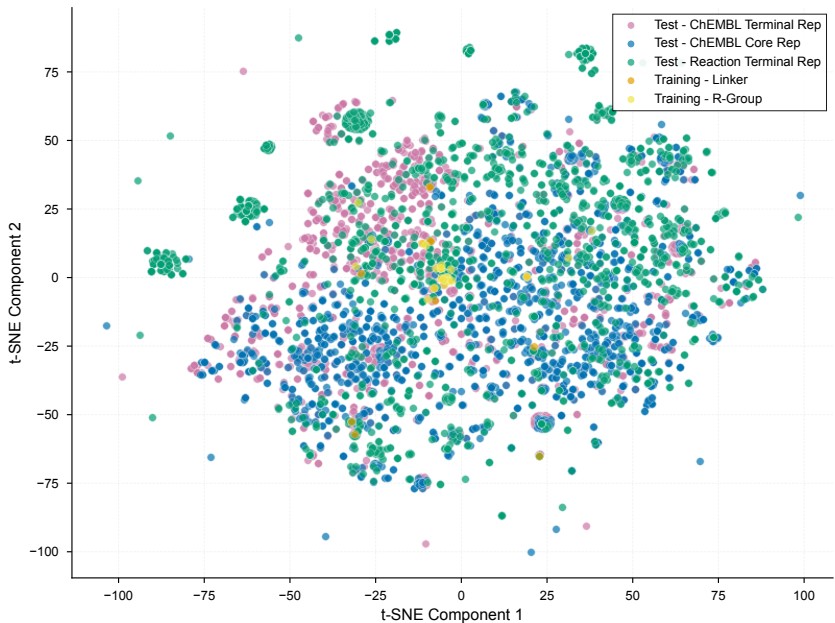

Figure 6: t-SNE for fragments in synthetic training data construction and realistic edit test set.

This spread confirms that Realistic Edits cover a substantially broader and more diverse chemical space. Despite this distribution shift, the finetuned code LLM in MECo maintains strong performance, supporting our claim that the model learns generalizable code transformation patterns rather than relying on fragment-level memorization.

## B MODEL TRAINING DETAILS

### B.1 PROMPT FORMAT AND DESIGN RATIONALE

**Prompt pre-filtering.** To determine an optimal prompting strategy, we conducted preliminary experiments examining the effects of various formatting choices. These preliminary experiments were conducted under conditions that differ from our final evaluation tasks; they serve only as pilot studies to guide the choice of attachment-point style.

For numbering attachment points in the SMILES, we tested:

- Mark attachment points only in source SMILES with atom map numbers;
- Numerically number all atoms in SMILES and give the corresponding numbers of attachment points.

For marking attachment points in the fragment, we wrote original and target fragments with:

- atom map number, e.g., [C:1]N
- rooted at attachment point with a short dash, e.g., -CN
- a asterisk mark, or a so-called dummy atom, e.g., *CN

Table 5: Accuracy across different SMILES attachment styles and numbering strategies.

| Action/Model | Molecule Number Style | | Attachment Point | Whole SMILES |
|---|---|---|---|---|
| | Fragment Attachment Style | | | |
| CodeGen / Coder | Numbered | [C:1]N | 0% | 0% |
| | Rooted | -CN | 20% | 1% |
| | Asterisked | *CN | 6% | 6% |
| Direct / General | Numbered | [C:1]N | 0% | 3% |
| | Rooted | -CN | 12% | 6% |
| | Asterisked | *CN | 4% | 4% |

Preliminary results are shown in Table 5. While the asterisk symbol ($*$) was not always the most accurate in pilot trials, it was retained because of its conventional role in representing dummy atoms in cheminformatics toolkits (e.g., RDKit, ChemDraw) and its compatibility with code generation. We also observed that passing numbered SMILES directly to the reasoning LLM leads to more stable responses, likely due to the limited power of LLMs handling SMILES during reasoning. In contrast, the rooted attachment-point style is not compatible with fragments requiring multiple attachment points, such as in core-replacement tasks.

---

**Final Prompt Format**

```
You are given a molecule in SMILES format:
"{numerically_numbered_source_smiles}".

For reference, atoms where new groups will be attached are marked with
"[*:n]", where "n" is the atom mapping number.

Future connected atoms in groups are labeled using the same numbers,
ensuring one-to-one attachment correspondence. You will then be given
multiple instructions on how to edit the molecule.

Replace the substructure corresponding to
"{original_fragment_smarts_1}"
connected at atom {numbers_indicating_attachment_points}
with "{target_fragment_smarts_1}".

...

Replace the substructure corresponding to
"{original_fragment_smarts_n}"
connected at atom {numbers_indicating_attachment_points}
with "{target_fragment_smarts_n}".

Generate a Python code snippet that performs these replacements
using RDKit via ChemicalReaction.

Ensure the code is executable and returns the modified molecule as
a new SMILES string.

You must return a Python code snippet wrapped in triple backticks.

The code should import modules from RDKit, perform the operation,
and print only the modified molecule in SMILES format.
```

## B.2 TRAINING SETUP

We used the official fine-tuning framework from the Qwen2.5-Coder repository[1] and conducted full-parameter supervised fine-tuning (SFT) using DeepSpeed on both `Qwen2.5-Coder-7B-Instruct` and `Qwen2.5-7B-Instruct` models. The training was performed on 4 NVIDIA A100 80GB GPUs, with each epoch taking approximately 4 hours.

## C EXPERIMENT DETAILS

### C.1 EXECUTION ACCURACY ACROSS FRAGMENT SIMILARITY BINS

To further assess whether MECo generalizes beyond the structural motifs present in its synthetic training set, we evaluate execution accuracy across fragment similarity bins computed using both structural similarity (ECFP–Tanimoto) and string similarity (SequenceMatcher ). Figure 7 reports accuracy curves over a wide range of similarity scores, covering multiple realistic edit test sources, including ChEMBL for activity-derived edits, USPTO for reaction-derived edits, and both terminal and core replacement types.

The results show that MECo maintains stable performance across the entire similarity spectrum, including bins where fragments are substantially dissimilar from those in the synthetic moiety pool. This provides further evidence that MECo's execution ability is not tied to neither structural coverage nor string pattern overlap of fragments in the training data, but instead generalizes robustly to structurally diverse and previously unseen edits.

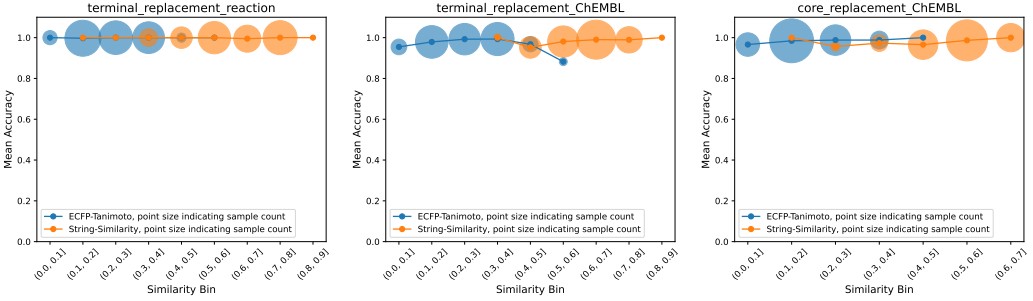

Figure 7: Execution accuracy of the finetuned code LLM (Qwen2.5-Coder-7B-Instruct-FT) across bins of structural similarity (ECFP–Tanimoto) and string similarity (SequenceMatcher), evaluated on data from different sources and for both terminal and core replacements. The size of the semi-transparent halo around each point reflects the number of test samples in that bin.

---

[1]https://github.com/QwenLM/Qwen2.5-Coder/tree/main/finetuning/sft

## C.2 PROMPT WRAPPER IN THE MOLECULAR OPTIMIZATION TASK

```
{Objective description}

Your response must be actions to perform the transformation, i.e.
"replace original_group_smiles connected at atom_number with
target_group_smiles".

Each action must include two SMILES strings indicating the original
group and the target group.

Use dummy atoms to mark the connection point, e.g.
"replace [*:1]Cl connected at atom_number with [*:1]OC"
or
"replace [*2:]c1ccc([*:3])cc1 connected at atom_number2, atom_number3
 with [*2:]c1cnc([*:3])cc1".

Your response must be in directly parsable JSON format:
{
    "Action Description": [
        "replace original_group_smiles connected

        at atom_number with target_group_smiles",

        ...
    ],
    "Final Target Molecule": "SMILES"
}

Given the source molecule with atoms numbered:
{number_smi(source_smiles)}.

You should ignore Hydrogen (H) and numbers in the group SMILES.
```

## C.3 CRITERIA AND EVALUATION FOR CONSISTENCY MANUAL CHECK

**Criteria.** We considered syntax errors in actions that are still discernible to human experts. Examples include malformed atom map annotations such as [*1:] or [*1], which should be [*:1], and missing explicit hydrogens on aromatic nitrogens, e.g., [*:1]c1nnnn1, which should be [*:1]c1[nH]nnn1.

For attachment point checks, we adopted relatively loose criteria. For instance, in [cH:1]1[cH:2][cH:3][cH:4][cH:5][c:6]1[O:7][CH2:8][CH:9]1[CH2:10][CH2:11]1, assigning the attachment of [*:1]OCC1CC1 to atom 6 or 7 is both considered discernible. Similarly, in [*:1]O[*:2], providing only one side of the atom index (e.g., 6 or 8) or the atom index alone (7) is also regarded as discernible. By contrast, edits that invoke groups not present in the structure are considered invalid, even if similar patterns can be found.

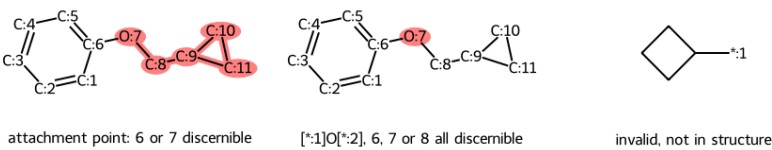

attachment point: 6 or 7 discernible          [*:1]O[*:2], 6, 7 or 8 all discernible          invalid, not in structure

Figure 8: Visualization of manual check criteria.

Nevertheless, manual checking is inherently subjective. Different researchers may interpret borderline cases differently, and there remains a possibility of human error.

**Subjectivity.** To further assess potential subjectivity, we adopt a blinded multi-annotator protocol: three annotators independently evaluate 120 samples (evenly drawn from DeepSeek-R1), without access to model identities, task provenance, or one another's labels. Structural labels (SMILES) were likewise highly consistent: three-way agreement = 95.83%, two-way = 3.33%, and complete disagreement = 0.83%. Attachment-point partiality judgments achieved Fleiss' $\kappa = 0.9342$ for the 104 jointly valid cases, indicating almost perfect agreement.

**Scalability.** This verification pipeline naturally supports scalable dataset growth: for each natural-language edit instruction, annotators provide one gold-standard structure (e.g., SMILES) that correctly implements the edit. Each labeled (instruction, targeted structure) pair becomes a reusable test case and can be automatically applied to any future code-LLM output via structure matching. Thus, each round of human labeling incrementally expands a standardized, reproducible, and low-cost evaluation suite. In addition, the annotation process can be substantially accelerated through formalized validation code. For example, automatically excluding substructure errors (e.g., fragments not present in the source molecule) before human inspection. This further reduces subjective load and ensures that annotators only review chemically valid candidate edits.

Regarding scalability toward larger and more diverse test sets, we draw on recent progress in LLM evaluation and outline several feasible extensions:

1. **LLM-as-a-Judge with minimal supervision.** Lin & Chen (2023); Li et al. (2025b) show that structured prompting enables LLMs to reliably approximate human judgments at scale, suggesting that MECo's intention-consistency checks could be extended through supervised automatic judges.

2. **Decomposed, verifiable evaluation dimensions.** Zhou et al. (2023); Ye et al. (2023) shows that breaking complex behaviors into verifiable atomic criteria yields more stable and interpretable assessments. Future MECo versions may evaluate edits along dimensions such as (i) substructure localization, (ii) bond/valence correctness, (iii) atom-mapping agreement, and (iv) chemically valid topology, paired with automatic graph/SMILES validators. This reduces subjectivity and aligns evaluation with chemically interpretable components.

3. **Standardized benchmark construction.** Successful evaluation suites (Zhou et al., 2023; Zheng et al., 2023) emphasize fixed task sets, unified templates, and executable scoring pipelines. Inspired by these designs, we could build a standardized benchmark for molecular edit–intention consistency with public task suites and unified evaluation scripts, ensuring fully reproducible cross-model comparison.

These extensions provide a clear and technically grounded path toward scaling MECo's intention-consistency evaluation beyond the manually assessed subset while enhancing reproducibility, transparency, and objectivity.

C.4 OBSERVATION ON TERMINAL REPLACEMENT TEST SAMPLES

In Section 4.2, we noted a discrepancy between reaction-derived and target-specific terminal replacements. To investigate this further, we conducted a closer inspection and found that the discrepancy may stem from differences in SMILES syntax patterns: in target-specific edits, dummy atoms (e.g., `[*:1]`) often appear in the middle or at the end of SMILES strings, whereas in reaction-derived edits, they frequently occur at the beginning. This difference arise from the implementation of the extraction program rather than the underlying data distribution, meaning that the two syntax types can be transformed into each other by choosing whether the dummy atom or another atom is used as the SMILES root. Such pattern sensitivity suggests that LLMs struggle to generalize over SMILES syntax, further motivating the use of structured code as a more robust and interpretable intermediate representation.

C.5 PROPERTY IMPROVEMENT VS. SIMILARITY

To further analyze the trade-off between property improvement and structural preservation, we compare the results of direct generation and MECo separately on all six tasks (Figure 9). The top row shows the average property improvement (relative to the initial molecule), where MECo generally achieves higher improvements than direct generation. The bottom row shows the average structural similarity (ECFP4-based) to the initial molecule. While Gemini-2.5-Pro exhibits larger improve-

ments in the upper row, it tends to reduce similarity more than DeepSeek-R1, indicating that different LLMs show different levels of willingness to sacrifice structural similarity in exchange for property gains. This observation motivates our use of success rate as the primary evaluation metric in the main paper, as it balances both optimization effectiveness and structural plausibility.

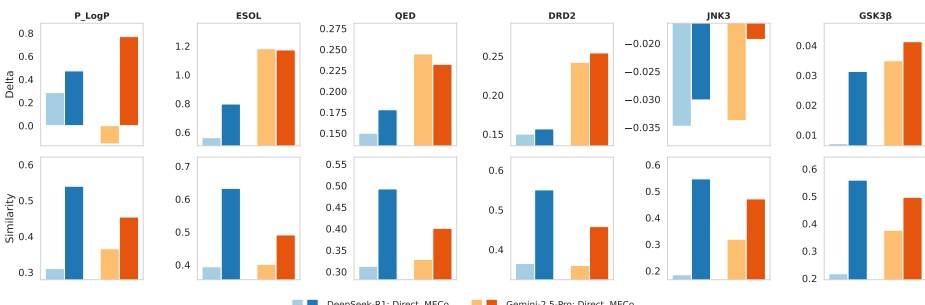

Figure 9: Bar plots of molecular optimization performance across six tasks, showing property improvement (top row) and structural similarity (bottom row) for Direct generation and MECo for DeepSeek-R1 and Gemini-2.5-Pro.

### C.6 DOCKING SETUPS FOR CASE STUDY

For the case study, molecular docking was performed using the Glide SP (standard precision) protocol implemented in the Schrödinger suite. The high-resolution X-ray co-crystallized target protein structure was obtained from the Protein Data Bank (PDB ID: 4AFJ).

Protein preparation was carried out using the Protein Preparation Wizard, including addition of hydrogen atoms, assignment of bond orders, optimization of the hydrogen-bonding network, and restrained minimization of heavy atoms with the OPLS_2005 force field. Ligands were prepared using LigPrep to generate at most 32 stereoisomers.

Docking was performed with default Glide SP parameters unless otherwise specified. The receptor grid was centered on the co-crystallized ligand SJJ in 4AFJ. The top-ranked docking poses were retained for analysis and visualization.

### C.7 COMPARISON WITH A GRAPH-BASED BASELINE

To contextualize MECo against a representative graph-generation approach, we performed a controlled comparison with HierG2G (Jin et al., 2020a). Since graph-based methods including HierG2G typically rely on oracle-driven supervision to learn the property landscape, whereas MECo operates purely in a zero-oracle regime, the experimental setup was adjusted to approximate a more comparable setting. For each target, we progressively removed HierG2G's original training samples similar to those in the test set, thereby inducing a "reduced-oracle" condition. In addition, HierG2G was trained using the same non-objective edit dataset employed for training the MECo code editor, which simulates the absence of target-specific optimization signals, i.e., a 0-oracle setting.

Under this setting, we evaluated both models on afore-mentioned metrics (Table 6) and on the structure–property Pareto frontier (Figure 10), using ECFP–Tanimoto similarity ($\geq 0.4$) as the structural constraint. On DRD2, HierG2G's optimization quality decreased as increasingly similar training samples were removed. MECo showed clear advantages, achieving higher improvements and a more favorable Pareto front, particularly in the high-similarity region. On QED, HierG2G exhibited weaker sensitivity to the removal of similar samples, likely due to the simplicity of the endpoint and the intrinsically lower similarity structure of the dataset.

These results suggest that graph-based generative models perform strongly when oracle-like supervision is readily available, but their performance degrades as such signals are removed. MECo, by relying on chemically informed reasoning rather than oracle-driven landscape exploration, provides more robust optimization capability in zero-oracle or low-information conditions. This underscores

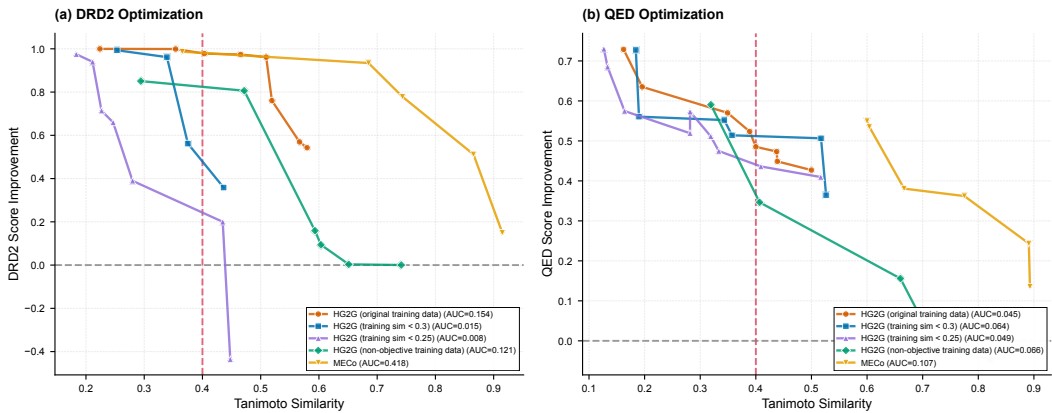

Figure 10: Pareto fronts illustrating the structure–property trade-off achieved by MECo and HierG2G on DRD2 and QED optimization tasks.

the complementary nature of the two approaches and motivates future combinations of language-driven reasoning with oracle/graph-based optimization frameworks.

Table 6: Comparison between HierG2G and MECo under varying training data configurations on DRD2 and QED optimization tasks. Equivalent oracle calls were calculated using positive rate reported in Jin et al. (2018b). Reported VR% (valid rate) counts only molecules that remain chemically valid and are successfully modified after optimization; molecules that are unchanged are considered invalid.

| Model / Condition | Training data size | Equivalent oracle calls | VR% | SR% | Improv. | Sim | Pareto AUC (%) |
|---|---|---|---|---|---|---|---|
| **DRD2** | | | | | | | |
| HierG2G (original training data) | 34,234 | 1,801,789 | 98 | 81 | 0.45 | 0.28 | 15.4 |
| HierG2G (training-to-test sim < 0.3) | 7,015 | 369,211 | 98 | 82 | 0.43 | 0.19 | 1.5 |
| HierG2G (training-to-test sim < 0.25) | 1,723 | 90,684 | 95 | 67 | 0.18 | 0.15 | 0.8 |
| HierG2G (non-objective training data) | 50,000 | 0 | 95 | 24 | −0.08 | 0.32 | 12.1 |
| MECo | 50,000 | 0 | 89 | 66 | 0.16 | 0.55 | 41.8 |
| **QED** | | | | | | | |
| HierG2G (original training data) | 87,226 | 1,321,606 | 77 | 77 | 0.33 | 0.19 | 4.5 |
| HierG2G (training-to-test sim < 0.3) | 64,886 | 983,121 | 77 | 77 | 0.32 | 0.20 | 6.4 |
| HierG2G (training-to-test sim < 0.25) | 37,196 | 563,576 | 77 | 77 | 0.32 | 0.17 | 4.9 |
| HierG2G (non-objective training data) | 50,000 | 0 | 92 | 31 | −0.06 | 0.30 | 6.6 |
| MECo | 50,000 | 0 | 87 | 75 | 0.18 | 0.49 | 10.7 |

# D  LLM USAGE STATEMENT

We employed LLMs in limited roles that do not affect the scientific claims of this paper. Specifically, reasoning LLMs and code LLMs were integrated as components of our proposed framework to generate molecular editing intentions and executable code, respectively. In addition, GPT-4o was used for assisting in editing the manuscript for grammar and clarity.

