# OpenReview forum: "Coder as Editor: Code‑driven Interpretable Molecular Optimization"
_ICLR.cc/2026/Conference — Submitted to ICLR 2026_

### Official Review · Reviewer_8VMz · 2025-10-25

**Soundness:** 2
**Presentation:** 2
**Contribution:** 3
**Rating:** 4
**Confidence:** 5

**Summary:**

This paper proposes MECo (Coder as Editor), a framework that reformulates molecular optimization as a code generation task. Instead of directly producing SMILES strings, it uses a reasoning LLM to propose interpretable editing intentions, and a coder LLM to translate them into executable RDKit scripts. Experiments on both synthetic and real benchmarks show that MECo significantly improves success and consistency rates across six property and activity optimization tasks. The approach demonstrates a promising paradigm for making LLM-based molecular design verifiable and reproducible rather than purely generative.

**Strengths:**

1. The code-based interface makes every molecular edit explicit, traceable, and executable, improving transparency and human understanding.

2. The framework consistently outperforms SMILES-based baselines across six property and activity optimization tasks.

**Weaknesses:**

## Motivation needs to be clarified with more evidence

1. In Abstract: The statement "LLMs have shown promise in generating high-level editing intentions in natural language" lacks supporting evidence. Is there any data in the paper or related work demonstrating this claim? Are there quantitative results (e.g., percentage, accuracy, or error rate) showing that the generated intentions (such as functional groups) are useful for a downstream task?

2. In Line 73, the authors skip a more straightforward approach, i.e., supervised fine-tuning (SFT). Why not simply construct molecular pairs before and after editing to evaluate the LLM’s performance in optimization? Another straightforward idea is to combine deterministic algorithms (such as MOLLEO, which the authors also discuss) or to apply multiple sampling with LLMs, which would be computationally inexpensive since each SMILES string is relatively short. If such simple methods were not used, an analysis explaining why and a comparison with them as baselines are needed.

## Details on the method are missing

1. The coder LLM edits molecules based on a set of predefined SMARTS patterns. However, the coverage and diversity of these patterns are unclear. A comprehensive analysis is needed to support the claimed generalization ability of the method.

2. In the coding process, one molecule may contain multiple substructures that match a given SMARTS pattern. How does the method decide which one to modify, or are all matching substructures replaced simultaneously? If the coder LLM writes the script and makes decisions, how do we know that the coder LLM strictly follows the instructions provided by the reasoning LLM?

## Related work and baseline

The work overlooks many previous efforts on inverse design prior to the use of LLMs and includes no non-LLM baselines. It is unclear why these earlier models are omitted, especially since there is no evidence showing that LLM-based molecular design outperforms non-LLM methods.

**Questions:**

1. All baselines are SMILES-based LLMs. Molecular generation constrained by structure has also been studied using graph-based models [1,2,3,4]. The work needs a more comprehensive comparison with these baselines.

2. The actions in the “reasoning” component are limited to predefined edit rules based on SMARTS. What if no suitable pattern from the predefined SMARTS rules exists for a given task?

3. What does "Qwen2.5-7B-Instruct-FT" mean in Table 1?

4. Why are there many "-" in Table 2?

5. Is there any complete case for the generation result of reasoning/coder LLMs?

Reference:
[1] Hierarchical Generation of Molecular Graphs using Structural Motifs. ICML 2020.
[2] Junction Tree Variational Autoencoder for Molecular Graph Generation. ICML 2018.
[3] Multi-Objective Molecule Generation using Interpretable Substructures. ICML 2020.
[4] DiGress: Discrete Denoising diffusion for graph generation. ICLR 2023.

---

> ### Author Response · Authors · 2025-11-21
> **Response to Weakness 1: Motivation and Evidence for MECo**
>
> ## W1. Motivation and evidence.
>
> We thank the reviewer for carefully examining the motivation. Our goal is to clarify that the motivation behind MECo is **not** the claim that LLMs already outperform on molecular optimization, but rather that LLMs enable a fundamentally different optimization paradigm, one based on **interpretable, human-auditable, and controllable editing intentions**, instead of direct end-to-end generation.
>
> Therefore, the motivation does not rely on presenting large-scale "data-level evidence" that LLMs already solve the task; instead, we argue that the ability to express intentions in natural language provides important advantages that previous models (GNNs, sequence-to-sequence generators, and diffusion models) cannot offer:
>
> 1.	Interpretability & Human-AI collaboration.
> Editing intentions are readable by chemists, allowing incorporation of medicinal-chemistry knowledge, expert critique, multi-step reasoning, and safety checks. This establishes a human-AI collaborative workflow that black-box model outputs cannot support.
>
> 2.	Explainability and traceability.
> The reasoning steps and the executable SMARTS edit scripts act as a self-explaining mechanism. Traditional paired-data translators cannot provide this level of transparency.
>
> 3.	Generalization beyond paired training regimes.
> Intent-based editing avoids the need for large paired low-to-high datasets (often costly or suffering from sim2real gap, e.g. a pre-trained predictor), and supports zero-shot editing on unseen chemical space.
>
> These are **design-level motivations**, not performance claims; thus they are not validated by a single percentage or accuracy number but by the design properties of the proposed framework.
>
> That said, we do include **empirical signals** supporting the usefulness of reasoning-driven intentions:
>
> * As shown in Table 2, reasoning LLMs (noted as *w/ thinking*, e.g., R1) generate more coherent and structurally consistent editing instructions compared to non-reasoning models (e.g., V3) under identical prompts, and lead to higher execution success rates.
> * Related work such as MolReasoner (Zhao et al., 2025) demonstrates that chain-of-thought-style reasoning benefits structured molecular optimization tasks.
>
> Together, these provide empirical evidence that structured reasoning enhances downstream editing quality.

---

> ### Author Response · Authors · 2025-11-21
> **Response to Weakness 1 (continued): Paired SFT and Deterministic Baselines**
>
> ## W1.2 Why not use paired SFT or deterministic algorithms as baselines?
>
> We thank the reviewer for raising this important point. We clarify that the suggested baselines, including paired-data SFT and deterministic search-based frameworks such as MOLLEO, operate under fundamentally different assumptions from our method.
>
> 1. **Paired-data SFT is not applicable because our setting is strictly oracle-free and pair-free.**
>
> 	Constructing molecular "before-after" editing pairs requires supervised labels, which in practice are obtained from:
>
> 	* oracle scoring, or
> 	* task-specific optimization trajectories (e.g., RL, MCTS, EA).
>
> 	Our method explicitly targets a **zero-shot setting where neither oracle scores nor optimization traces are available**.
> 	Training SFT on paired data would defeat the core purpose of MECo, i.e., learning edit intention generation **without any task-specific supervision**.
>
> 	Thus, paired-data SFT for molecular optimization is incompatible with our experimental setting, and including it as a baseline would be an unfair comparison because it accesses supervision that our method does not use.
>
> 	We did include a paired-data SFT baseline in **Table 1** ("Qwen2.5-7B-Instruct-FT") fine-tuned on editing-to-molecule execution, which is a general LLM and **underperforms the fine-tuned code LLM** used in the MECo framework.
>
> 2. **Deterministic algorithms such as MOLLEO rely on repeated oracle queries, again a different problem setting.**
>
> 	The reviewer mentioned MOLLEO as a "straightforward" deterministic algorithm. However:
>
> 	* MOLLEO repeatedly calls an LLM and repeatedly calls the oracle to evaluate intermediate molecules.
> 	* This forms an iterative optimization loop (initial → LLM modify → oracle → loop).
> 	* MECo instead generates both (a) a consistent edit intention and (b) an optimized molecule in a **single forward pass, without oracle access**.
>
> 	Therefore, MOLLEO-type methods solve a guided optimization problem, while MECo solves a single-shot oracle-free editing problem.
> 	A direct comparison would conflate two tasks with different supervision and computational requirements.
>
> 3. **Multiple-sampling LLM baselines are orthogonal and can be combined with MECo.**
>
> 	Sampling the LLM many times to increase diversity is not an edit-coherence mechanism.
> 	Our contribution is **orthogonal**:
>
> 	* MECo enforces consistency between the natural-language edit intention and the generated molecule within a single inference, improving modification quality.
> 	* Multi-sampling only increases the number of candidates, but requires multiple oracle calls, which can be costly when relying on experimental validation rather than a predictor.
>
> 	For fairness, all our LLM baselines use the same number of forward passes (one), aligning with the oracle-free single-shot setting.
> 	We highlight that MECo can actually be plugged into frameworks like MOLLEO or multi-sampling, since it solves a different challenge: **intent-structure alignment in one pass**.
>
> The suggested baselines rely on additional supervision (paired data) or additional computational budget (iterative oracle calls) or solve a different task (guided optimization rather than oracle-free editing).
> Our work focuses on a **minimal-supervision, zero-oracle, single-inference setting**, which is increasingly **important for real-world early-stage design without accessible property oracles**.

---

> ### Author Response · Authors · 2025-11-21
> **Response to Weakness 2: Method Details and Generalization Beyond SMARTS Patterns**
>
> ## W2. Details on the method are missing
>
> ### W2.1 Analysis supporting the claimed generalization ability
>
> We appreciate the reviewer's concern regarding coverage and diversity of the predefined SMARTS patterns. Our analysis is expanded to clarify how MECo generalizes beyond the fragments used in training.
>
> 1. **The evaluation set explicitly excludes fragments similar to those used in synthetic training**.
> For the Realistic Edits test set, we de-duplicate all examples whose pre-/post-edit fragments have an ECFP-based Tanimoto similarity ≥ 0.6 to any fragment in the synthetic moiety pool. This ensures that MECo is evaluated on fragment types not seen during training.
>
> 2. **The Realistic Edits set occupies a substantially broader chemical space**.
> We added an ECFP-based t-SNE visualization (Appendix A.5 & Figure 6), showing that fragments from Realistic Edits are far more diverse and widely distributed than the synthetic pool. This indicates that the test distribution is deliberately challenging and not covered by the predefined SMARTS patterns.
>
> 3. **MECo's generalization stems from learning transformation patterns, not memorizing fragment types**.
> Unlike enumerative SMARTS-rule systems, MECo does not rely on expanding the training set to cover every fragment. The code LLM learns structural transformation patterns in Python/SMARTS logic, e.g., how to identify the edited subgraph and preserve the attachment topology, rather than fragment-specific chemistry. Once these transformation patterns are learned, unseen fragments are naturally handled as long as they instantiate the learned structural relationships.
>
> 4. **Performance remains stable across structural and string similarity bins**.
> To further validate this, we added an analysis curve (Appendix C.1 & Figure 7) showing that MECo's accuracy is stable across the whole range of fragment similarity bins (both ECFP-Tanimoto and SequenceMatcher), covering multiple sources (ChEMBL activity edits, USPTO reaction edits) and both terminal and core replacements. This supports that MECo maintains consistent performance even when the fragments differ significantly from the training distribution.
>
> ### W2.2 Simultaneous occurrence issue
>
> We thank the reviewer for raising this question. MECo avoids ambiguity in cases where multiple substructures match the same SMARTS pattern by **explicitly defining both the fragment pattern and its attachment point(s)** in the prompt to the coder LLM.
>
> 1. **Unique identification of the edited substructure.**
> As described in Appendix B.1 (Prompt format and design rationale), the reasoning LLM outputs not only the substructure SMARTS but also the indices of the attachment atom(s). This paired SMARTS pattern and atom indices uniquely identify the target subgraph even when multiple matches exist. If several matches are structurally equivalent, replacing any one yields the same chemical edit, so the choice does not affect correctness.
>
> 2. **Design alignment across training, coding, and evaluation.**
> This design is consistent across all stages:
>
> 	* Training data: Algorithm 1 in Appendix A.1 records the exact atom indices of the fragment selected for replacement.
> 	* Downstream optimization: The reasoning LLM must specify edits using the same "pattern + indices" convention.
> 	* Realistic test data:
> 		* Reaction-based edits use atom-mapping to determine the edited atoms.
> 		* ChEMBL activity edits derive fragment-to-replace and its remnant part explicitly.
>
> 3. **Thus, the coder LLM performs no decision-making.**
> Under this setup, the coder LLM simply executes the transformation specified by the reasoning LLM. It does not choose among multiple candidates or infer the edit position.
>
> 4. **Consistency is empirically evaluated.**
> We quantify how faithfully the coder LLM executes the instructions using the Consistency Rate (CR%) reported in Table 2. This directly measures whether the generated SMILES reflect exactly the intended subgraph replacement.

---

> ### Author Response · Authors · 2025-11-21
> **Response to Weakness 3 and Question 1: Graph-Based Baselines in a Zero-Oracle Setting**
>
> ## W3 & Q1. Absence of Non-LLM / Graph-Based Constrained Generation Baselines
>
> We thank the reviewer for emphasizing the importance of non-LLM and graph-based constrained generation methods for molecular optimization. Before going into setting differences, we would like to clarify the **contribution and scope** of MECo in relation to these methods. MECo is not proposed as a new generator, but as a **code-based, interpretable editing framework** that maps intelligible, reasoning-based edit intentions to deterministic molecular transformations. Its main contribution is to strengthen **intent–structure alignment** and human auditability, in a way that is complementary to existing graph and diffusion-based molecular optimization methods.
>
> Within this perspective, we agree that non-LLM and graph-based constrained generation approaches are highly relevant and have made significant contributions to structure-constrained generation.
>
> Our work, however, focuses on a different but complementary capability: **zero-oracle, interpretable molecular editing driven by LLM reasoning**. During optimization, MECo and all LLM baselines generate edit intentions and edited molecules in a **single forward pass, without querying any property or activity oracle** and without relying on paired low-to-high optimization trajectories. In contrast, classical graph and diffusion-based models typically assume access to oracle scores (or dense supervision) during training and/or optimization, and therefore operate in a different assumption space.
>
> To bridge this gap and better connect with prior work, we have added a representative graph-based baseline, HierG2G [1], adapted to a low-/zero-oracle regime. As detailed in **Appendix C.7 (Figure 10, Table 6)**, we:
>
> 1. Progressively prune HierG2G's training data by enforcing low similarity between training and test molecules, and
> 2. Evaluate a non-objective variant trained only on general edit-style data (0 equivalent oracle calls), analogous to MECo's zero-oracle setting.
>
> Under these configurations, HierG2G's optimization performance degrades markedly, while MECo remains effective. For example, in the non-objective (0-oracle) setting, HierG2G achieves 24% success rate and 12.1% Pareto AUC on DRD2, whereas **MECo attains 66% success rate and 41.8% Pareto AUC**. On QED, non-objective HierG2G yields 31% success rate with negative mean improvement and 6.6% Pareto AUC, while **MECo reaches 75% success rate and 10.7% Pareto AUC**.
>
> In the revised manuscript, we have **expanded Section 2 (Related Work)** to discuss representative graph-based and diffusion-based constrained generation models and to clearly explain that MECo is intended to be **complementary**: graph models excel when rich oracle feedback is available, whereas MECo aims to unlock LLM-based, human-auditable reasoning for **zero-oracle or early-stage design scenarios**.
>
> [1] Hierarchical Generation of Molecular Graphs using Structural Motifs. ICML 2020.

---

> ### Author Response · Authors · 2025-11-21
> **Response to Questions 2–5: Reasoning Rules, Table Notation, and Case Study Details**
>
> ## Q2. Limited predefined edit rules in reasoning.
>
> We clarify that **the reasoning LLM is not constrained by any predefined edit rules** (Appendix C.2 (original C.1) shows the prompt). The SMARTS-based edit rules are used only for constructing synthetic training data for the **coder LLM**, whose role is to execute explicit subgraph replacements deterministically. These rules are not used to guide or restrict the reasoning LLM's high-level intentions.
>
> Instead, as detailed in reponse to **Weakness 2.1**, the realistic test set is carefully de-duplicated to remove any edit whose before/after fragments resemble the synthetic training moieties (Tanimoto ≥ 0.6). The t-SNE analysis further shows that realistic edits occupy a much broader chemical space than the synthetic rules. MECo succeeds on these realistic edits without expanding the predefined patterns, and the accuracy is irrelevant to neither structural nor string similarity to the predefined set, suggesting that the reasoning LLM would not rely on them.
>
> Thus, predefined edit rules are **solely a training setting for the coder LLM, not a limitation placed on the reasoning component**.
>
> ## Q3. Meaning of Qwen2.5-7B-Instruct-FT in Table 1
>
> `Qwen2.5-7B-Instruct-FT` is a baseline where we finetune the general-purpose LLM `Qwen2.5-7B-Instruct` on **paired SMILES (source to target)** to directly generate the optimized SMILES from the same prompt.
> In contrast, our approach trains a code-specialized LLM that performs explicit, localized edits using SMARTS-based transformations.
>
> The substantially lower accuracy of Qwen2.5-7B-Instruct-FT highlights that:
>
> 1.	Direct SMILES-to-SMILES generation struggles to preserve structural constraints and execute precise edits.
> 2.	Code-based editing is far more sample-efficient and robust for executing edit instructions.
>
> This comparison underscores the benefit of learning structured edit transformations in the format of code snippets rather than relying on end-to-end SMILES generation for edit execution.
>
> ## Q4. "-" in Table 2
>
> The symbol "-" appears in the **CR% (consistency rate)** column denotes the consistency between the reasoning LLM's edit proposal and the code LLM's execution.
> For models with substantially lower performance, we did not report CR% for the two following reasons:
>
> 1.	Non-reasoning LLMs frequently fail to produce valid structured edit descriptions, making consistency evaluation ill-posed.
> 2.	GPT-5 occasionally declines to answer due to its internal safety striction, preventing us from obtaining evaluable outputs.
>
> Thus, "-" indicates that the **consistency evaluation was not applicable or was not conducted**.
>
> ## Q5. Complete case for the generation result of reasoning/coder LLMs
>
> We provide the complete generation record corresponding to the case shown in Figure 4.B in the Supplementary Material, including all prompts and intermediate outputs from both the reasoning and coder LLMs.

---

### Official Review · Reviewer_3vVR · 2025-10-28

**Soundness:** 3
**Presentation:** 3
**Contribution:** 2
**Rating:** 6
**Confidence:** 3

**Summary:**

The paper proposes MECo, a framework that reformulates molecular optimization as a code-generation problem. Instead of directly generating SMILES strings, a reasoning LLM first produces high-level editing intentions, and a separate coder LLM translates these intentions into executable scripts that perform the structural edits.

**Strengths:**

1. Reformulating molecular editing as a code generation problem is an original and practical idea.
2. Each generated edit is represented as an explicit Python/RDKit script, which allows human chemists to read, check, and reproduce the exact transformation.
3. The model demonstrates that LLMs can indeed produce executable and syntactically correct chemical code.

**Weaknesses:**

1. The framework is only compared with general-purpose LLMs, not with specialized molecular optimization models, which makes its competitive value for drug discovery unclear.
2. The system appears to follow a linear pipeline: the reasoning LLM generates an edit, and then the executor applies it. However, this design underutilizes the LLM’s reflection ability. In the future, the authors could consider allowing the reasoning LLM to reflect on the executor’s results and perform multi-round optimizations.

**Questions:**

see weakness

---

> ### Author Response · Authors · 2025-11-21
> **Response to Weaknesses 1 & 2: Positioning of MECo and Single-Round Pipeline Design**
>
> ## W1. Comparison Only with General-Purpose LLMs Instead of Specialized Molecular Optimization Models
>
> We thank the reviewer for raising this important point. Before discussing settings, we would like to clarify the **contribution and scope** of MECo in relation to specialized molecular optimization models. MECo is not intended to be a new generator or a substitute for oracle-driven optimization pipelines; rather, it provides a **code-based, interpretable editing framework** that translates high-level edit intentions into precise RDKit transformations. The core contribution is to strengthen **intent–structure alignment** and human-auditable editing, which can sit alongside many existing optimization back-ends.
>
> Within this framing, we fully agree that specialized graph-based and reaction-based molecular optimization models are strong baselines in traditional oracle-driven settings.
>
> Our work, however, is designed for a different regime: MECo and all LLM baselines operate in a **strict zero-oracle setting**, where neither the reasoning LLM nor the code LLM ever query activity/property oracles or use paired low-to-high optimization data. Instead, MECo leverages the internal chemical knowledge of reasoning LLMs to propose **interpretable edit intentions**, and uses a code LLM to deterministically execute these edits as RDKit code.
>
> By contrast, specialized optimization models such as JT-VAE, GraphAF, hierarchical graph generators, and diffusion-based methods typically rely on repeated oracle calls or dense supervised trajectories to shape their optimization behavior. They are therefore not directly comparable under our zero-oracle, single-shot assumption, and including them as main baselines would mix fundamentally different problem settings.
>
> Nevertheless, to strengthen the empirical context, we have added a representative graph baseline, HierG2G [1], adapted to a comparable low-/zero-oracle condition. In **Appendix C.7 (Figure 10, Table 6)**, we show that when HierG2G is trained without task-specific objective data (0 equivalent oracle calls), its optimization performance deteriorates sharply, whereas **MECo maintains 66% success rate and 41.8% Pareto AUC on DRD2, and 75% success rate and 10.7% Pareto AUC on QED**. These results suggest that MECo remains **competitive, and often stronger, in the strict zero-oracle regime** where LLM reasoning is particularly valuable.
>
> We have also **expanded Section 2 (Related Work)** in the revised manuscript to explicitly discuss these specialized models and to clarify that MECo is intended to be **complementary**: in oracle-rich settings, MECo could be integrated as a reasoning-and-editing front-end within classical optimization pipelines, while our current work focuses on demonstrating its effectiveness in the more constrained, oracle-free scenario.
>
> [1] Hierarchical Generation of Molecular Graphs using Structural Motifs. ICML 2020.
>
> ## W2. Linear Pipeline and Underuse of LLM Reflection / Multi-Round Optimization
>
> We appreciate the reviewer's insightful comment. The current work indeed adopts a linear pipeline, in which the reasoning LLM proposes an edit and the coder LLM executes it once. This design is intentional: our primary goal is to **isolate and evaluate the benefit of code-based, interpretable editing**, without conflating it with agent-style multi-round optimization strategies.
>
> That said, MECo is naturally compatible with reflection and multi-round optimization. Because the edit intentions are explicit and the code-based edits are deterministic, one can readily form a loop of
>
> **reasoning LLM → code LLM → execution → analysis of results → next-round reasoning**,
>
> potentially incorporating additional feedback such as property predictions, uncertainty estimates, or human-in-the-loop critique.
>
> We view our current single-round formulation as a **initial, well-controlled step** that demonstrates the value of interpretable reasoning-based editing. Building on this foundation, we plan to explore reflective and multi-round variants of MECo in future work. We have clarified this extensibility in **Section 5 (Conclusion)**, where we explicitly mention agent-style extensions and multi-round optimization as a promising direction for future work, so that the connection is clear to readers.

---

### Official Review · Reviewer_zr88 · 2025-10-31

**Soundness:** 3
**Presentation:** 4
**Contribution:** 3
**Rating:** 4
**Confidence:** 5

**Summary:**

The paper proposes MECo (Molecular Editing via Code generation), a cascaded framework for molecular optimization: a reasoning LLM generates human-interpretable editing actions, and a coder LLM translates these actions into executable RDKit code that edits the input molecule. A synthetic training set of moiety/linker substitutions is used to fine-tune a code model; evaluation uses three “realistic” edit sets (reaction-derived, target-specific terminal, target-specific core) and ChemCoTBench optimization tasks. MECo reports ~98% edit execution accuracy on realistic edits and higher success/consistency on downstream optimization versus direct SMILES generation by LLMs.

**Strengths:**

- Clear problem decomposition. Separating “why edit” (reasoning) from “how to edit” (code execution) addresses a known brittleness of SMILES-level generation.
- Strong execution fidelity. The code LLM fine-tuned on synthetic edits achieves >=98% accuracy on all three realistic edit categories, a large margin over direct SMILES generation and zero-shot code prompting baselines.
- Meaningful end-task gains. On ChemCoTBench, MECo improves success rate and similarity while reaching ~90–98% structure–action consistency, which is important for traceable design.
- Reproducible ingredients. RDKit-based edits, explicit prompt wrapper, and synthetic data construction details improve reusability.
- Case studies with mechanistic narrative. The examples show plausible edits that retain cores and avoid score hacking behaviors seen in direct generation.

**Weaknesses:**

- Reliance on synthetic-to-real transfer without ablations. The model is trained only on synthetic replacements yet attains ~98% on realistic edits; stronger leakage checks and ablations (e.g., reducing pattern overlap, altering SMARTS syntax, training set size scaling) are not reported. Filtering based on Tanimoto >= 0.6 to training moieties may be insufficient to exclude near-variants of patterns.
- Manual consistency scoring introduces subjectivity. Structure–action consistency depends on a manual protocol that accepts “discernible” but syntactically imperfect actions. No inter-rater agreement, adjudication process, or blinded evaluation is reported.
- Baseline breadth and parity. Comparisons focus on direct LLM generation and simple code prompting. Strong graph-editing or reaction-transform baselines (e.g., JT-VAE/graph AF variants configured for localized edits, template-guided transformations) are not evaluated.
- Metric choices can favor conservative edits. Emphasis on success rate and similarity, with mean property improvement deferred to the appendix, may bias toward small edits. A multi-objective view (e.g., Pareto fronts of improvement vs. similarity) is not shown.

**Questions:**

- How robust is MECo to stereochemistry and formal charge preservation? Please report chirality retention rates and valence error checks after edits.
- Does ChemicalReaction editing ever produce multiple products or atom-mapping ambiguities? How are products filtered and how often?
- What is the impact of training set size and pattern diversity on execution accuracy? Please add scaling-law style ablations.

---

> ### Author Response · Authors · 2025-11-21
> **Response to Weakness 1: Generalization Beyond Synthetic Patterns**
>
> ## W1. Analysis supporting generalization beyond synthetic patterns
>
> We appreciate the reviewer's concern regarding potential leakage from synthetic SMARTS-based patterns into the Realistic Edits evaluation. We have expanded the analysis to directly address synthetic-to-real transfer, fragment coverage, and whether MECo truly generalizes beyond the predefined patterns.
>
> 1. **The Realistic Edits test set covers a substantially broader chemical space than the synthetic pattern pool.**
>
> 	We provide a new ECFP-based **t-SNE visualization (Appendix A.5, Figure 6)**, showing that Realistic Edits fragments occupy a much wider and more heterogeneous region of chemical space than the synthetic moiety pool. This indicates that the real-world evaluation distribution is significantly shifted and cannot be reconstructed by trivial variants of the synthetic patterns.
>
> 	Strict leakage controls are enforced at the subgraph level.
> 	Beyond simple fragment similarity filtering, the synthetic training patterns and realistic evaluation fragments differ in:
>
>    * their complexity: tens of synthetic enumerations are further less to cover the vast varity of Realistic edits,
>    * their structural motifs and atom types: many aromatic heterocycles or non-organic/metal atoms appear only in Realistic Edits.
>
> 	This subgraph/topology difference cannot be captured by SMARTS syntax overlap alone and effectively prevents pattern memorization from explaining the performance.
>
> 2. **Fragment-level similarity does not explain MECo's high accuracy.**
>
> 	To examine potential leakage or near-duplicate effects, we added an accuracy curve **across both structural similarity (ECFP-Tanimoto) and string similarity (SequenceMatcher) bins (Appendix C.1, Figure 7)**.
> 	**Across the entire similarity spectrum, MECo maintains stable execution accuracy.**
> 	This strongly suggests that performance is not driven by exposure to similar SMARTS patterns, but by genuine generalization to unseen fragment structures.
>
> 	MECo generalizes by learning transformation code, not fragment patterns.
> 	The code LLM does not memorize fragments; instead, it uses procedural transformation patterns expressed in Python/SMARTS (e.g., locating the subgraph defined by SMARTS patterns and attachment indices, replacing it using well-established `ReactionFromSmarts` function).
> 	Once these transformation rules are learned, new fragments are naturally handled as long as they fit the learned structural mapping pattern, regardless of their atom types or substitution patterns.
> 	This mechanism differs fundamentally from direct manipulation on SMILES string level, which require explicit coverage of each new fragment type.
>
> Together, these analyses demonstrate that MECo's high accuracy on realistic edits is not caused by leakage but by genuine generalization beyond the synthetic SMARTS patterns used for training.

---

> ### Author Response · Authors · 2025-11-21
> **Response to Weakness 2: Human Consistency Scoring and Subjectivity**
>
> ## W2. Subjectivity of Manual Consistency Scoring and Lack of Agreement Statistics
>
> We thank the reviewer for raising this concern. In MECo, human evaluation is restricted to a clearly defined notion of edit-intention consistency, as described in the revised **Appendix C.3** and Figure 8.
>
> To reduce subjectivity and quantify reliability, we adopt a **blinded multi-annotator protocol**: three chemistry-literate annotators independently label each sample without knowing the model identity, task provenance, or each other's decisions. Across 120 evaluated samples (evenly drawn from DeepSeek-R1), action validity reached 95.83% three-way agreement (115/120 samples with jointly identical canonical SMILES or jointly invalid). For the 104 jointly valid cases, attachment-point judgments achieved Fleiss' kappa = 0.9342, indicating almost perfect agreement. Structural labels (SMILES) were likewise highly consistent, with only 0.83% complete disagreement; these rare cases are resolved by a brief post-hoc adjudication, consistently yielding a single unambiguous interpretation.
>
> Regarding scalability, our verification pipeline naturally supports dataset accumulation and reuse: for each natural-language edit instruction, annotators provide one gold-standard structure (e.g., SMILES) that correctly implements the edit. Each (instruction, gold structure) pair then becomes a reusable test case that can be automatically matched against outputs from any future model, enabling large-scale, low-cost evaluation without repeated manual inspection. The process can be further accelerated by formalized validation code (e.g., automatically discarding samples with fragments unmatching the source molecule) so that annotators only review chemically valid edits.
>
> Finally, we discuss in the revised appendix C.3 how recent LLM-as-a-judge and decomposed-evaluation frameworks can be adapted to further automate intention-consistency scoring while maintaining interpretability. Together, we hope the additional evaluation results and planned extensions address the reviewer's concerns about subjectivity, reproducibility, and scalability of manual consistency checks.

---

> ### Author Response · Authors · 2025-11-21
> **Response to Weaknesses 3 & 4: Baseline and Evaluation Metrics**
>
> ## W3. Missing Graph-Editing / Reaction-Transform Baselines and Baseline Parity
>
> Before turning to specific baseline choices, we would like to clarify the **contribution and scope** of MECo in this work. MECo is not proposed as a new generator, but as a **code-based, interpretable editing framework** that connects natural-language edit intentions to deterministic molecular transformations. The main goal is to improve **intent–structure alignment** and human auditability in reasoning-based molecular editing, and to **complement rather than replace** existing oracle-based optimizers, whether graph- or reaction-based.
>
> Within this scope, we fully acknowledge that graph-based and template-guided optimization models are highly relevant and effective in traditional, oracle-driven molecular optimization.
>
> Our experimental setting, however, is deliberately constrained to a **zero-oracle, single-shot optimization regime**. MECo and all LLM baselines generate both the edit intention and the optimized molecule in a single forward pass, **without calling any property or activity oracle** during proposal. In contrast, classical graph-editing and reaction-based frameworks (e.g., JT-VAE, GraphAF variants, template-guided transformations) are typically trained and deployed with extensive oracle guidance (or dense paired low-to-high data), and are therefore not directly comparable under the same assumptions.
>
> To better connect with this literature, we have added a representative graph-generative baseline, HierG2G [1], adapted to a low-/zero-oracle regime. Specifically, in **Appendix C.7 (Figure 10, Table 6)** we:
>
> 1. Progressively filter HierG2G's training molecules to enforce low similarity to the test set (training-to-test similarity <0.3, then <0.25), and
> 2. Evaluate a non-objective variant trained only on general edit-style data (without task-specific objectives), which corresponds to **0 equivalent oracle calls**, matching MECo's zero-oracle condition.
>
> Under these configurations, HierG2G's optimization performance degrades substantially, while MECo remains robust. For example:
>
> - On the DRD2 task, the non-objective HierG2G variant (0-oracle) achieves 24% success rate and 12.1% Pareto AUC, whereas **MECo (also 0-oracle) reaches 66% success rate and 41.8% Pareto AUC**.
> - On the QED task, the non-objective HierG2G variant yields 31% success rate with negative mean improvement and 6.6% Pareto AUC, while **MECo achieves 75% success rate and 10.7% Pareto AUC**.
>
> These results indicate that when graph-based models are deprived of oracle-derived supervision, their optimization ability sharply declines, whereas MECo continues to provide meaningful improvements by leveraging LLM reasoning and code-based execution. We have **expanded Section 2 (Related Work)** in the revised manuscript to explicitly discuss these differences in assumption space and to position MECo as complementary to oracle-driven graph and reaction-transform baselines.
>
> [1] Hierarchical Generation of Molecular Graphs using Structural Motifs. ICML 2020.
>
> ## W4. metric choices and "conservativeness"
>
> We respectfully clarify that structural similarity is not a bias of our evaluation protocol, but a defining requirement of molecular optimization. In practical design settings, **maintaining a reasonable degree of structural continuity is essential for interpretability, synthetic feasibility, and experimental validation**. Extremely low-similarity edits, although they may produce molecules with high property values, typically reflect retrieval or memorization rather than meaningful optimization, and fall outside the intended scope of the task.
>
> Importantly, we **do not impose any preference for conservative edits in our prompts**, which are fully transparent (both in Appendix B.1 & C.2 and ChemCoTBench). The different behavior observed across reasoning LLMs arises from their intrinsic editing tendencies rather than from our evaluation design:
>
> * DeepSeek-R1 tends to propose smaller edits yielding higher-similarity, moderate-improvement.
> * Gemini-2.5-pro produces lower-similarity, higher-magnitude changes.
>
> This divergence highlights model-specific optimization strategies, not a bias in our metrics. This tendency **can be guided by oriented prompt if needed**.
>
> We also note that mean property improvement, provided in the Appendix C.5 and Figure 9, shows that **MECo consistently delivers steady improvements across tasks** while perserves realistic similarity constraints. This indicates that strong optimization does not require extreme or unconstrained modifications.
>
> We agree that multi-objective metrics such as **Pareto fronts** are valuable in certain optimization settings, and provide such analyses in **Appendix C.7 and Figure 10**.

---

> ### Author Response · Authors · 2025-11-21
> **Response to Questions 1-3: Chemical Validity and SFT Ablations**
>
> ## Q1. stereochemistry and formal charge preservation, valence check
>
> MECo executes modifications through RDKit's `ReactionFromSmarts.RunReactants` operator, without directly manipulating SMILES strings, molecular graphs or atom coordinates using neural networks. In this framework, **stereochemistry, formal charges and connectivities are preserved automatically by RDKit**: atoms outside the edited substructure retain their original chirality and charge because these attributes are stored on RDKit's `Atom` objects within the `Mol` object, which can not be violated by MECo.
>
> **Valence validity is also guaranteed by design.** Any valence error causes RDKit `sanitize` to fail, in which case no SMILES is produced. Thus, every successful MECo output necessarily corresponds to a molecule with chemically valid valence.
>
> Given these mechanism-level guarantees enforced deterministically by RDKit, we believe additional quantitative metrics for stereochemistry preservation or valence validity are unlikely to provide further practical value in this context.
>
> ## Q2. Handling of Multiple Products and Atom-Mapping Ambiguities in ChemicalReaction Edits
>
> We thank the reviewer for raising this important point. We ensure unambiguous atom mapping through the design: Substructure SMARTS and attachment point atom indices (canonicalized via RDKit) **guarantee unambiguous atom mapping**, which is consistent across code LLM training (Algorithm 1, Appendix A.1), realistic edit tests (code LLM prompt, Appendix B.1), and molecular optimization tasks (reasoning LLM prompt, Appendix C.2).
>
> Multiple products usually arise from symmetry in substructure matches and are equivalent. Selecting the first one suffices in most cases.
>
> Partial attachment point annotations (e.g., marking only one end of a multi-attachment fragment or neighboring atoms inside the fragment) are still localized correctly using containment relationships.
> We performed a robust product selection by filtering **matches both the source molecule's residual part (after fragment removal) and the target fragment**, ensuring correct final products.
>
> Empirical statistics across six molecular optimization tasks show:
>
> * Multiple equivalent products occur in 24-45% of reactions (overall ~33%).
> * Multiple distinct products occur in 5-22% of reactions (overall ~12%) and are reliably filtered using the containment-based approach.
>
> These results demonstrate that ambiguities are rare, mostly equivalent, and handled robustly by the generated code.
>
> ## Q3. The impact of training set size and pattern diversity on execution accuracy
>
> We thank the reviewer for this suggestion. The reviewer's request appears to relate LLM scaling laws to our SFT ablation, but these two are essentially unrelated. Scaling laws describe how model capacity and emergent reasoning abilities evolve with pretraining compute, data, and parameter count (e.g., $10^8-10^{12}$ tokens). In contrast, our experiment concerns a small supervised fine-tuning stage (50k samples) on top of an already strong code LLM, where no emergent behaviors are expected and classical scaling-law phenomena do not apply.
>
> Here, the key question is not about expanding model capacity, but about whether the SFT data provides sufficient coverage of the required transformation patterns. While the fragment pattern space is small (fragment and attachment patterns are limited to tens of enumerated forms), reducing the training set size substantially reduces the diversity of source molecular patterns encountered during fine-tuning.
>
> Empirically, the results in the table below confirm this:
>
> * Core replacement accuracy remains essentially unchanged from 1k to 50k samples.
> * Terminal replacement accuracy is already very high (>97%) and shows only marginal improvement with larger datasets.
>
> These results indicate that execution accuracy is largely insensitive to training-set size or source molecular pattern diversity. The SFT stage mainly aligns the model's coding behavior to the molecular editing task, rather than teaching new chemical knowledge or depending on specific fragment or molecular patterns.
>
> ||data size |1,025|4,100|16,400|50,000|
> |---|---|---|---|---|---|
> |Terminal replacement|Reaction-derived|99.7|99.6|99.7|99.9|
> | |Target-specific|97.8|96.8|98.2|98.3|
> |Core replacement|Target-specific|98.5|98.4|98.8|98.3|

---

> > ### Comment · Reviewer_zr88 · 2025-11-26
> >
> > I appreciate your responses to my questions and concerns. I have updated my score accordingly. Some additional content you clarified above should be included in the final manuscript.

---

> > > ### Author Response · Authors · 2025-11-26
> > >
> > > Thank you for the reassessment and positive update. We will integrate the clarified points into the final manuscript.

---

### Official Review · Reviewer_xmoc · 2025-10-31

**Soundness:** 3
**Presentation:** 3
**Contribution:** 2
**Rating:** 4
**Confidence:** 4

**Summary:**

This paper introduces MECo, a framework for interpretable molecular optimization that leverages large language models (LLMs) to generate high-level molecular editing intentions in natural language, which are then translated into executable code for structural modification of molecules. MECo decouples chemical reasoning (intent formation) from execution (precise code-driven edits), utilizing a code LLM (Qwen2.5-Coder) to translate rationales into robust RDKit-based editing scripts. The authors present a systematic synthetic and realistic data construction pipeline for training and evaluation. MECo demonstrates strong accuracy on edit execution benchmarks and achieves higher success, similarity, and intention-consistency rates than SMILES-based generation and previous LLM approaches on molecular optimization tasks.

**Strengths:**

1. The MECo framework proposes a well-motivated separation between intent (reasoning LLM) and edit execution (code LLM), resulting in interpretable and robust molecular modifications. This mirrors expert workflows and addresses a long-standing bottleneck in LLM-driven molecule design (Section 1, Figure 1).

2. The evaluation is comprehensive, involving both synthetic and realistic molecular edits, with clear and challenging test splits (see Section 3.3, Figure 2B). Filtering out test samples with high similarity to training moieties ensures a genuine assessment of model generalization.

3. MECo achieves impressive execution accuracy ($98%+$) on realistic edits (Table 1) and consistently higher performance (success rate, similarity, and edit-consistency) across multiple property and bioactivity optimization tasks (Table 2, Figure 3).

4. The code-based approach produces auditable and modifiable editing scripts, greatly aiding transparency and reproducibility. Figure 4B demonstrates how intention, code, and edit outcomes remain aligned in complex cases.
Careful Metrics & Human Evaluation: The authors avoid misleading optimization metrics (see discussion in Section 3.5, Appendix C.4) by focusing on success rates and consistency, and they clearly describe robust manual criteria for edit intention realization (Appendix C.2, Figure 5).

**Weaknesses:**

1. Several highly relevant recent works on code-driven molecular optimization and code LLM reasoning, such as Yu et al. (2025) (Collaborative Expert LLMs Guided Multi-Objective Molecular Optimization), are not cited or compared. These works also address LLM-guided optimization with structured intermediates and could expose theoretical or empirical overlaps.
Influential studies on LLM code analysis (Fang et al., 2024), and on splitting code generation between LLMs and formal program synthesis (Murphy et al., 2024), are omitted from the related work and could further contextualize the challenges and limits of code-driven approaches in molecular tasks (see Section 2, missing from the discussion).

2. While Table 2 covers a range of LLM-based approaches, the comparison almost exclusively focuses on direct SMILES generation or "reasoning LLM + direct execution" without including strong recent graph-based or programmatic editing methods as empirical baselines (e.g., advanced molecular graph editing, or hybrid neuro-symbolic models).
No dedicated ablation of the prompt format choices or an investigation into how prompting and code granularity impacts robustness and execution success, especially in edge or ambiguous editing scenarios (prompting decisions are described in Section B.1, but not experimentally dissected).

3. The reliance on DeepSeek-R1 (and to some extent Gemini-2.5-Pro) as the intention LLM leaves unclear how the overall system behaves under less capable or non-scientific LLMs, and whether the overall gains are primarily due to the code LLM, reasoning LLM, or their synergy (Section 4.3, Table 2). There is some analysis of improvements across reasoning LLMs (Figure 3), but a more detailed breakdown would strengthen claims of generality.

4. For multi-site (core) replacements and edits involving complex attachment point mapping, the formal definition of permutation handling (Appendix A.4) is described, but the main paper does not fully explore the algorithmic consequences (e.g., how ambiguities in fragment symmetry or atom mapping are resolved in executable code). This could affect robustness in less-structured or fuzzy design scenarios.

5. Human-in-the-loop evaluations (see Section 4.3, Appendix C.2 & Figure 5) are clear and motivated, but reliance on manual consistency checks for intention realization raises concerns of scalability, subjectivity, and reproducibility on very large or diverse benchmarks.

6. While the formalism for code generation is largely clear (Section 3.3, Algorithm 1), the mechanism for fragment identification, attachment point matching, and the translation of ambiguous instructions into unambiguous, executable RDKit code is not fully formalized. For example, in iterative moiety replacement (Algorithm 1), how does the model (or supporting code) resolve cases where multiple matching substructures are present, or when multiple attachment permutations are valid? These details are nontrivial and could be potential failure modes.

7. Furthermore, in Table 1, some discrepancies between reaction-derived and target-specific replacements are discussed (Appendix C.3), but specific SMILES syntax challenges (such as dummy atom placement) and code-generation edge-cases are not quantitatively analyzed in detail.

8. Although the authors claim strong generalization from synthetic to realistic edits, the training data’s moiety/replacement set is well-defined (Table 3 and 4), and it remains an open question whether broader, less formulaic edits—such as those arising from less constrained drug design—can be captured without an explosion of training diversity.

**Questions:**

1. In cases where the same edit action (natural language) could map to multiple code realizations (e.g., due to substructure symmetry or ambiguous atom mapping), how does MECo determine which to prioritize, and what are the observed failure rates as a function of edit ambiguity? Explicit discussion of error distributions and representative code translation mistakes would be welcome.
2. Can the manual edit-intention consistency checks be reliably scaled to order-of-magnitude larger or more diverse test sets, or is there a feasible path toward automating this evaluation in a way that preserves interpretability and trust?
3. What is the impact on MECo’s downstream success and consistency if a less performant or non-scientifically trained reasoning LLM is used? Have the authors observed any bottlenecks or cascading failures rooted in the upstream intent-generation stage, and how sensitive is MECo to noisy or partially incorrect edit actions?
4. Is the code-based editing approach extensible to larger classes of modifications, such as multi-step or scaffold-hopping edits, or to more open-ended design objectives (e.g., multi-objective, combinatorial libraries)? What (if any) bottlenecks emerge in code complexity, execution speed, or model training for such cases?

---

> ### Author Response · Authors · 2025-11-21
> **Response to Weaknesses 1 & 2: Positioning of MECo and Baseline Coverage**
>
> ## W1. Missing Related Work on Code-Driven Optimization
>
> We thank the reviewer for pointing this out. We have added **discussion in Section 2 and Section 3.1 (paragraph "Why code as an interface?")** covering recent molecular optimization and code generation works (Yu et al., 2025; Fang et al., 2024; Murphy et al., 2024). MECo focuses on aligning high-level molecular editing intentions with precise code execution, enabling interpretable edits that can support multi-agent coordination and benefit from advances in code generation. These additions better contextualize the scope and concept of MECo.
>
> ## W2. Limited Non-LLM / Graph Baselines and Prompt Ablation
>
> Before discussing specific baselines, we would like to clarify the **contribution and scope** of MECo in the context of molecular optimization. MECo is not a new generator or a drop-in replacement for existing oracle-driven optimization pipelines. Instead, it introduces a **code-based, interpretable editing framework** that aligns high-level, natural-language edit intentions with precise, executable molecular transformations. The central contribution is to **bridge intent and structure** through LLM reasoning plus code execution, enabling human-auditable, controllable edits that can plug into different optimization regimes.
>
> Within this scope, we fully agree that classical graph-editing and reaction-transform models are powerful in oracle-driven optimization settings, and we do not claim to supersede them in their native, oracle-rich regimes.
>
> Our work, however, targets a **strict zero-oracle and human-interpretable optimization regime**. During proposal and reasoning, MECo and all LLM baselines **never query any activity or property oracle**, and do not rely on paired low-to-high optimization data. In contrast, classical graph-based optimization frameworks (e.g., JT-VAE, GraphAF, hierarchical or diffusion-based generators) are typically trained and deployed with extensive oracle feedback or dense paired data, and thus cannot be fairly evaluated under our zero-oracle setting.
>
> To address the reviewer's concern, we have nevertheless added a representative graph-editing baseline, HierG2G [1], adapted to a comparable low-/zero-oracle regime. In **Appendix C.7 (Figure 10, Table 6)**, we progressively reduce the similarity between HierG2G's training and test molecules and also evaluate a non-objective variant trained only on general edit-style data (the same editor-only data used for MECo's coder). Under this configuration, HierG2G's performance deteriorates sharply, while MECo remains effective:
>
> - On DRD2, when trained without task-specific objective data ("non-objective" condition, **0 oracle calls**), HierG2G achieves **24%** success rate and **12.1%** Pareto AUC, whereas **MECo (also 0-oracle) reaches 66% success rate and 41.8% Pareto AUC**.
> - On QED, the non-objective HierG2G baseline again loses optimization ability (success rate 31%, negative mean improvement), while **MECo maintains 75% success rate and improves Pareto AUC from 6.6% to 10.7%**.
>
> These results show that when graph-based models are deprived of oracle-derived signals, their optimization ability degrades, whereas MECo, by leveraging LLM reasoning and code-based execution, maintains robust performance in the zero-oracle regime. We have also **expanded Section 2 (Related Work)** to better position MECo as complementary to classical graph-based and reaction-based optimization methods, rather than as a direct replacement.
>
> Regarding prompt ablations and code granularity, we have added additional qualitative and quantitative analysis in **Appendix B.1 and Table 5**. We compare alternative prompt formats and code templates, and observe that our final design offers a good balance between robustness, compatibility with cheminformatics toolkits, and the ability to handle multi-attachment operations such as core replacement and scaffold hopping.
>
> [1] Hierarchical Generation of Molecular Graphs using Structural Motifs. ICML 2020.

---

> ### Author Response · Authors · 2025-11-21
> **Response to Weaknesses 3 & 4 and Question3: System Components, Robustness, and Ambiguity Handling**
>
> ## W3/Q3. Sensitivity to Reasoning Model Quality and Overall Gains Breakdown
>
> We thank the reviewer for the valuable comment. Indeed, the MECo system relies on a capable reasoning LLM to generate meaningful molecular modifications and translate them into structured descriptions. Less capable models often produce unrealistic SMILES and fail to generate parseable modification descriptions due to weaker domain knowledge, chemical language, and instruction-following ability. Even when part of their descriptions can be parsed by the code LLM, the underlying edit proposals are inherently less reasonable and effective. As a result, integrating such weaker models into the MECo framework does not yield stable or consistent gains in property improvement, although MECo still helps ensure **structural similarity and correctness** of the resulting SMILES.
>
> We would like to clarify that MECo's molecular optimization capability primarily derives from the reasoning LLM's chemical knowledge and reasoning ability. However, the reasoning LLM alone cannot fully realize this potential due to limitations in SMILES-level execution. The code LLM serves as a complement rather than an enhancer, faithfully executing the insights from the reasoning LLM. As shown in **Table 2**, the comparison between MECo and individual reasoning LLMs (DeepSeek-R1 / Gemini-2.5-Pro) demonstrates the **synergy of the system**. The code LLM alone, by design, does not perform molecular optimization, but ensures precise and robust execution of the reasoning LLM's intended edits.
>
> ## W4. Ambiguities in Multi-Site Replacement and Permutation Handling
>
> We thank the reviewer for raising the concern about "less-structured or fuzzy design scenarios." We would like to clarify that inputs to the code LLM are, by design, **specific and structured editing instructions**, ensuring that modifications remain well-defined even when multiple occurrences of a given substructure exist. In practice, the combination of attachment indices and substructure patterns defines a unique molecular subset in the vast majority of cases (100% in terminal replacement, and 92.5% in our sampled core replacement test set. As exemplified in **Appendix A.4 and new Figure 5**, Cases where ambiguity could arise (symmetry-to-asymmetry) are rare and can be addressed with more rigorous modification descriptions and post-filtering code if needed. However, imposing overly strict constraints can negatively impact model performance, and in scenarios such as physicochemical property optimization or when structural context is limited, the relative benefits of such constraints are often uncertain. Therefore, we enumerate all such possible permutations and treat each as correct in the core-replacement test, since MECo's robustness is not materially affected in molecular optimization tasks.

---

> ### Author Response · Authors · 2025-11-21
> **Response to Weakness 5 and Question 2: Human Evaluation Protocol and Scalability**
>
> ## W5/Q2. Scalability, Subjectivity, and Reproducibility of Manual Consistency Checks
>
> We thank the reviewer for highlighting concerns about the scalability and reproducibility of manual intention-realization checks. In **MECo**, human evaluation is restricted to a clearly defined set of criteria (**Appendix C.3 & Figure 8**).
>
> To further quantify subjectivity, we use a **blinded multi-annotator protocol**: three annotators are unaware of model identity, task provenance, or each other's labels. Across 120 evaluated samples (evenly drawn from DeepSeek-R1), action validity reached **95.83% agreement** (115/120 jointly identical canonical SMILES or jointly invalid). For the 104 jointly valid cases, attachment-point judgments achieved Fleiss' Kappa = 0.9342, indicating almost perfect agreement. Structural labels (SMILES) were likewise highly consistent: three-way agreement = 95.83%, two-way = 3.33%, and complete disagreement = 0.83%. The few disagreement cases are trivially resolved via seperate, post-hoc validation, consistently yielding a single unambiguous interpretation.
>
> ### Scalability of Manual Consistency Checks
>
> Our verification pipeline naturally supports scalable dataset growth: for each natural-language edit instruction, annotators provide one gold-standard structure (e.g., SMILES) that correctly implements the edit. Each labeled (instruction, targeted structure) pair becomes a **reusable test case** and can be automatically applied to any future code-LLM output via structure matching. Thus, each round of human labeling incrementally expands a standardized, reproducible, and low-cost evaluation suite. In addition, the annotation process can be substantially accelerated through **formalized validation code**. For example, automatically excluding substructure errors (e.g., fragments not present in the source molecule) before human inspection. This further reduces subjective load and ensures that annotators only review chemically valid candidate edits.
>
> ### Toward Automated and Interpretable Evaluation
>
> We see three feasible paths to scale MECo's intention-consistency evaluation while preserving interpretability and trust:
>
> 1. **LLM-as-a-Judge with minimal supervision.**
>    Recent systems [1,2] show that structured prompting enables LLMs to reliably approximate human judgments at scale, suggesting that MECo's intention-consistency checks could be extended through supervised automatic judges.
>
> 2. **Decomposed, verifiable evaluation dimensions.**
>    Research [3,4] shows that breaking complex behaviors into verifiable atomic criteria yields more stable and interpretable assessments. Future MECo versions may evaluate edits along dimensions such as (i) substructure localization, (ii) bond/valence correctness, (iii) atom-mapping agreement, and (iv) chemically valid topology, paired with automatic graph/SMILES validators. This reduces subjectivity and aligns evaluation with chemically interpretable components.
>
> 3. **Standardized benchmark construction.**
>    Successful evaluation suites [3,5] emphasize fixed task sets, unified templates, and executable scoring pipelines. Inspired by these designs, we could build a standardized benchmark for molecular edit-intention consistency with public task suites and unified evaluation scripts, ensuring fully reproducible cross-model comparison.
>
> These extensions provide a clear and technically grounded path toward scaling MECo's intention-consistency evaluation beyond the manually assessed subset while enhancing reproducibility, transparency, and objectivity. We thank the reviewer again for the constructive feedback.
>
> ### References
>
> [1] Lin, Y.-T. & Chen, Y.-N. *LLM-Eval: Unified Multi-Dimensional Automatic Evaluation for Open-Domain Conversations with Large Language Models.* arXiv:2305.13711 (2023).
> [2] Li, M., Li, H. & Tan, C. *HypoEval: Hypothesis-Guided Evaluation for Natural Language Generation.* arXiv:2504.07174 (2025).
> [3] Zhou, J. et al. *Instruction-Following Evaluation for Large Language Models.* arXiv:2311.07911 (2023).
> [4] Ye, S. et al. *FLASK: Fine-Grained Language Model Evaluation Based on Alignment Skill Sets.* arXiv:2307.10928 (2023).
> [5] Zheng, L. et al. *Judging LLM-as-a-Judge with MT-Bench and Chatbot Arena.* NeurIPS 36, 46595-46623 (2023).

---

> ### Author Response · Authors · 2025-11-21
> **Response to Weaknesses 6 & 7 and Question 1: Code-Level Execution Details and Error Analysis**
>
> ## W6/Q1. Fragment Identification, Attachment-Point Matching & Ambiguity Resolution
>
> We thank the reviewer for pointing out that the mechanism for fragment identification, attachment point matching, and translating ambiguous instructions into unambiguous, executable code could benefit from further clarification. We would first like to emphasize that Algorithm 1 is not a formal code-generation procedure, but rather a data-construction algorithm used to synthesize supervised training examples (source molecule, fragment, attachment-point annotations, and the corresponding target) for the code LLM (and general LLM as a baseline). The execution-time behavior, including matching sites location, and multiple product selection, is performed by the generated RDKit code, not by Algorithm 1.
>
> We would like to clarify the following regarding Algorithm 1:
>
> - Fragment identification is implemented via RDKit's `GetSubstructMatches` API, which deterministically returns atom indices of substructure matches in the target molecule.
> - Once the fragment is selected from all possible matches, the attachment-point matching is resolved by our numbering scheme: the reasoning LLM produces structured modification instructions (see Appendix C.2) with explicit attachment indices, thereby avoiding ambiguous natural-language references.
>
> We would like to clarify the following regarding the mechanism:
>
> - MECo aims to perform precise modifications based on reasoning outputs, rather than generating ambiguous instructions and attempting to resolve missing information (e.g., choosing among multiple matches). The instructions from reasoning LLMs are structured actions combining natural language and SMARTS (not free-form natural language intentions), and are **by design unambiguous**, directly translatable into deterministic molecular transformations and unique molecules. Consequently, the code LLM receives a unique mapping from these instructions and is **not responsible for resolving reasoning ambiguities**, but instead faithfully executes the provided mapping.
> - In cases with multiple matching substructures, our numbering system ensures a unique canonical match (unless chemically equivalent duplicates exist, in which case either mapping is acceptable).
> - Cases with multiple attachment permutations only arise under symmetry-to-asymmetry mappings, which comprise ~7.5% of our core-replacement test set, and do not materially affect robustness in molecular optimization tasks (see W4).
> - Accordingly, we believe that these details do not constitute practical failure modes under realistic use-cases of MECo.
>
> We have **updated algorithm 1** accordingly and provided a complete case for the generation result corresponding to Figure 4.B in the Supplementary Material to ensure transparency of the workflow and to address possible concerns of ambiguity or nondeterminism.
>
> ## W7. Quantitative Analysis of Code Translation Errors
>
> We thank the reviewer for raising this point. The discrepancy between reaction-derived and target-specific replacements is indeed an interesting observation, but it lies outside the core scope of our work. We note that such differences primarily appear when (i) directly generating SMILES strings or (ii) generating code without finetuning. In contrast, **within our framework, the effect is minimal**: even on the Realistic Edits dataset, which did not rule out unintended corner cases, the performance gap is only 1.6%, and the impact on molecular optimization endpoints is even more marginal.
>
> For this reason, we did not pursue a deeper quantitative breakdown. Instead, we highlighted this discrepancy as a possible starting point for future investigations into how LLMs understanding chemistry-specific languages such as SMARTS.
>
> We also appreciate the reviewer's suggestion regarding systematic analysis of code-generation edge cases. This is **highly aligned with our future plan** to extend MECo into agent workflows and engineering settings. At present, however, the code-generation reliability is already sufficient for the benchmark tasks and does not constitute the main bottleneck in our system.

---

> ### Author Response · Authors · 2025-11-21
> **Response to Weakness 8 and Question 4: Generalization and Extensibility of MECo**
>
> ## W8/Q4. Generalization Beyond Well-Defined Edits / Extensibility to Multi-Step or Scaffold-Hopping Edits
>
> Regarding the concern about generalization beyond the synthetic moiety-replacement training set:
>
> 1.	**Fragment diversity and generalization.**
> For realistic edits, we explicitly de-duplicated the test set by removing all examples whose pre-/post-edit fragments have similarity >= 0.6 to the fragments used for constructing the synthetic training data. As shown in the newly added ECFP-based t-SNE plot in the Appendix A.5, Figure 6, the fragments in the Realistic Edits set span a substantially broader and more diverse chemical space than the synthetic moiety pool. Importantly, MECo does not require expanding the training set to cover more fragment types or tasks: the code LLM learns patterns of code transformations, not fragment-specific chemistry. Once the mapping pattern is learned, new fragments are handled naturally.
> 2.	**Expressiveness of replacement operations.**
> From an editing-operation perspective, combinations of replacement operations—including addition (from nothing to a fragment) and deletion (from a fragment to nothing)—are expressive enough to transform any molecular graph into another. In downstream molecular optimization tasks, the reasoning LLM frequently proposes multi-site replacements, demonstrating that MECo can already execute complex edits beyond simple one-step changes.
> 3. **less constrained drug design?**
> If the reviewer is referring to extremely large or unconstrained modifications, we note that such edits are generally not aligned with the goals or practical constraints of molecular optimization. From both a controllability perspective and an experimental feasibility standpoint, modifications involving many (>3) concurrent structural changes effectively amount to de novo design rather than optimization, and are rarely used in realistic drug-design workflows. MECo focuses on edits that remain actionable and experimentally tractable, which is consistent with common medicinal chemistry practice.
> 4.	**Scaffold hopping.**
> The core-replacement task directly instantiates scaffold-hopping edits (see Fig. 2B). In downstream optimization, the reasoning LLM often performs scaffold-level proposals as well (see Fig. 4). Therefore, the system has already been tested on modifications substantially broader than local moiety substitutions.
> 5.	**Multi-step and iterative edits.**
> MECo naturally supports multi-step modifications. The synthetic data construction itself uses a multi-step loop (Appendix A.1 Algorithm 1), and in downstream tasks, the reasoning LLM outputs multi-site edits in a single pass, which the code LLM executes iteratively. For multi-round design, the reasoning → code-generation → execution cycle can simply be repeated. No architectural changes are required.
> 6.	**Multi-objective or open-ended design.**
> MECo's role is to reliably translate reasoning outputs into precise molecular edits. Multi-objective or combinatorial design affects the reasoning component, not the code-execution mechanism. As long as the reasoning LLM produces structured editing instructions, the code LLM can execute them without any additional training burden or complexity.
>
> If this does not fully address your question, we would appreciate clarification on what specific forms of "less constrained drug design" the reviewer has in mind, so we can respond more precisely.

---

### Author Response · Authors · 2025-12-03
**Summary of reviews and rebuttals**

We thank the reviewers for their constructive feedback. They appreciated MECo's novelty in reformulating molecular optimization as a reasoning-then-code generation problem. While reviewers raised concerns regarding **baseline comparisons** and **code LLM generalization/ambiguity**, we have fundamentally addressed these issues through new experiments and clarifications, as summarized below.

## 1. Reviewers' recognition of strengths

*   **Novelty & soundness:** Reviewers praised the "well-motivated separation between intent (reasoning LLM) and edit execution (code LLM)" (xmoc) and the "original and practical idea" of code-based editing (3vVR). Reviewer zr88 noted the "clear problem decomposition" addresses the brittleness of SMILES-level generation.
*   **Strong performance:** Reviewers acknowledged MECo's "impressive execution accuracy" (xmoc) and "strong execution fidelity" (~98% on realistic edits) (zr88), which consistently outperforms SMILES-based baselines (8VMz).
*   **Interpretability:** The code-based interface makes edits "explicit, traceable, and executable" (8VMz) and "auditable" (xmoc). Reviewer zr88 commended the "reproducible ingredients" and "mechanistic narrative" of the case studies.
*   **Rigorous evaluation:** Reviewer xmoc noted the "comprehensive" evaluation and "careful metrics" that avoid misleading optimization scores.

## 2. Addressed concerns

Multiple reviewers raised concerns across four key areas. Notably, Reviewer zr88, who **raised questions in all these areas**, has appreciated our responses and **raised their score**. We believe addressing these shared concerns resolves the reservations behind the initial borderline ratings.

### A. Lack of graph-based / non-LLM baselines (8VMz, 3vVR, zr88, xmoc)

**Concern:** Reviewers requested comparisons with specialized graph-based or reaction-based optimization models (e.g., HierG2G, JT-VAE) to benchmark MECo against non-LLM approaches.

**Response:**

*   **Clarified scope:** MECo operates in a **strict zero-oracle setting** (no objective function used during generation), unlike traditional models relying on oracle feedback.
*   **New baseline:** We adapted a representative graph-generative model, **HierG2G**, to the zero-oracle regime.
*   **Result:** In **Appendix C.7**, we show that without oracle guidance, HierG2G's performance degrades sharply (e.g., 24% success on DRD2), while **MECo remains robust (66% success)**. This demonstrates MECo's superior capability in early-stage, zero-oracle design scenarios.

### B. Generalization and potential data leakage (8VMz, zr88, xmoc)

**Concern:** Reviewers questioned whether the high performance on "Realistic Edits" was due to leakage from the synthetic training data or limited to predefined SMARTS patterns.

**Response:**

*   **Analysis:** We added an **ECFP-based t-SNE visualization (Appendix A.5)** showing that the realistic test fragments cover a much broader and distinct chemical space than the synthetic training pool.
*   **Leakage disproof:** We provided an accuracy curve across structural and string similarity bins to the training set (**Appendix C.1**), showing MECo maintains stable high accuracy across all similarity ranges.
*   **Clarification on pattern scope:** We clarified (addressing xmoc and 8VMz) that the patterns in the Appendix are solely for synthetic data generation. The code LLM learns general subgraph replacement rules, generalizing to "less constrained" designs and diverse chemical spaces beyond the training set.

### C. Ambiguity in edit execution (8VMz, zr88, xmoc)

**Concern:** Reviewers asked how the system handles cases where multiple substructures match a pattern or where atom mapping is ambiguous.

**Response:**

*   **Clarification:** Our prompt requires the reasoning LLM to output **explicit attachment atom indices** alongside SMARTS patterns.
*   **Role of code LLM:** The code LLM **faithfully executes** precise instructions. Ambiguity is resolved upstream by the Reasoning LLM's structured output, ensuring the code LLM does not make implicit structural decisions.

### D. Subjectivity of manual evaluation (zr88, xmoc)

**Concern:** Reviewers raised concerns about the scalability and subjectivity of the manual "intention consistency" checks.

**Response:**

*   **Protocol:** We detailed our **blinded multi-annotator protocol** (3 independent experts).
*   **High agreement:** We reported 95.83% inter-rater agreement on action validity and Fleiss' Kappa > 0.93 for attachment points, confirming the evaluation is objective and reliable.
*   **Scalability:** Verified (instruction, molecule) pairs become reusable automated test cases for future evaluations.

## Conclusion

We believe MECo represents a significant step forward in making molecular optimization interpretable and controllable. We have further solidified the paper's contribution by addressing the reviewers' concerns, and we are confident that the revised manuscript meets the high standards of ICLR.

---

### Meta-Review · Area_Chair_pZUJ · 2026-01-06

**Summary:**

This paper presents interesting ideas and has received borderline scores. After carefully reviewing both the reviews and the rebuttal, I find that several critical concerns have not been fully addressed, and therefore my recommendation is rejection. Reviewers xmoc and 8VMz raised a number of valuable points, which I strongly encourage the authors to consider seriously and address thoroughly in future revisions.

**Reviewer Concerns:**

Reviewer xmoc:

- W1. Missing Related Work on Code-Driven Optimization. Addressed.
- W2. Limited Non-LLM / Graph Baselines and Prompt Ablation. Not addressed. The authors reframe the problem by restricting the evaluation to a self-defined zero-oracle setting and evaluating graph baselines under degraded conditions, rather than providing competitive non-LLM baselines and prompt ablations in a standard, broadly accepted optimization regime.
- W3/Q3. Sensitivity to Reasoning Model Quality and Overall Gains Breakdown. Not fully addressed. The author's lack of quantitative sensitivity analysis and a clear empirical breakdown of gains means a reviewer could reasonably judge W3/Q3 as not fully addressed.
- W4. Ambiguities in Multi-Site Replacement and Permutation Handling. Partially addressed. The authors resolve the ambiguity concern by redefining ambiguous multi-site outcomes as all correct and relying on empirical rarity rather than providing a principled or general mechanism for disambiguation.
- W5/Q2. Scalability, Subjectivity, and Reproducibility of Manual Consistency Checks. Fully addressed.
-  W6/Q1. Fragment Identification, Attachment-Point Matching & Ambiguity Resolution. Partially addressed. The authors assume that ambiguity can be avoided by construction (structured instructions and numbering) rather than demonstrating a general, formally guaranteed resolution mechanism or quantitative failure analysis for fragment/attachment ambiguities at execution time.
- W7. Quantitative Analysis of Code Translation Errors. Not addressed. This is an important question, which should be addressed directly in the rebuttal.

---

Reviewer zr88 is satisfied with the rebuttal and raised the score to 6.

---

Reviewer 3vVR's score is 6.

---

Reviewer 8VMz

- W1. Motivation and evidence. Not addressed. Most support is conceptual, and the empirical signals cited (Table 2, MolReasoner) are indirect and do not directly validate the claimed advantages over prior optimization paradigms. The logic is broken here.
- W1.2 Why not use paired SFT or deterministic algorithms as baselines? Addressed.
- W2.1 Analysis supporting the claimed generalization ability. Addressed.
- W2.2 Simultaneous occurrence issue. Addressed.
- W3 & Q1. Absence of Non-LLM / Graph-Based Constrained Generation Baselines. A similar comment from reviewer xmoc. The authors address the absence of non-LLM/graph baselines mainly by redefining the problem to a self-imposed zero-oracle regime and evaluating graph methods under degraded conditions, rather than providing competitive constrained-generation baselines in their standard or broadly accepted settings.
- Q2 - Q5. Addressed.

**Reviewer Scores:**

The reviewers are likely to keep their scores.

---

### Decision · Program_Chairs · 2026-01-26

Reject